# Reward-Directed Conditional Diffusion: Provable Distribution Estimation and Reward Improvement

**Hui Yuan**     **Kaixuan Huang**

**Chengzhuo Ni**     **Minshuo Chen**     **Mengdi Wang** *

## Abstract

We explore the methodology and theory of reward-directed generation via conditional diffusion models. Directed generation aims to generate samples with desired properties as measured by a reward function, which has broad applications in generative AI, reinforcement learning, and computational biology. We consider the common learning scenario where the dataset consists of majorly unlabeled data and a small set of data with noisy reward labels. Our approach leverages a learned reward function on the smaller data set as a pseudolabeler to label the unlabelled data. After pseudo-labelling, a conditional diffusion model (CDM) is trained on the data and samples are generated by setting a target value $a$ as the condition in CDM. From a theoretical standpoint, we show that this directed generator can effectively learn and sample from the reward-conditioned data distribution: 1. our model is capable of recovering the data's latent subspace representation. 2. the model generates samples moving closer to the user-specified target. The improvement in rewards of samples is influenced by a interplay between the strength of the reward signal, the distribution shift, and the cost of off-support extrapolation. We provide empirical results to validate our theory and highlight the relationship between the strength of extrapolation and the quality of generated samples. Our code is available at https://github.com/Kaffaljidhmah2/RCGDM.

## 1  Introduction

Controlling the behavior of generative models towards desired properties is a major problem for deploying deep learning models for real-world usage. As large and powerful pre-trained generative models achieve steady improvements over the years, one increasingly important question is how to adopt generative models to fit the needs of a specific domain and to ensure the generation results satisfying certain constraints (e.g., safety, fairness, physical constraints) without sabotaging the power of the original pre-trained model [35, 27, 54, 41].

In this paper, we focus on directing the generation of diffusion models [19, 43], a family of score-matching generative models that have demonstrated the state-of-the-art performances in various domains, such as image generation [39, 38, 4] and audio generation, with fascinating potentials in broader domains, including text modeling [3, 27], reinforcement learning [21, 1, 36, 28] and protein structure modeling [26]. Diffusion models are trained to predict a clean version of the noised input, and generate data by sequentially removing noises and trying to find a cleaner version of the input. The denoising network (a.k.a. score network) $s(x,t)$ approximates the score function $\nabla \log p_t(x)$ [45, 46], and controls the behavior of diffusion models. People can incorporate any control information $y$ as an additional input to $s(x,y,t)$ during the training and inference [38, 54].

---

*Department of Electrical and Computer Engineering, Princeton University. Authors' emails are: {huiyuan, kaixuanh, cn10, mc0750, mengdiw}@princeton.edu

37th Conference on Neural Information Processing Systems (NeurIPS 2023).

Here we abstract various control goals as a scalar reward $y$, measuring how well the generated instance satisfies our desired properties. In this way, the directed generation problem becomes finding plausible instances with higher rewards and can be tackled via reward-conditioned diffusion models. The subtlety of this problem lies in that the two goals potentially conflict with each other: diffusion models are learned to generate instances *similar to* the training distribution, while maximizing the rewards of the generation drives the model to *deviate from* the training distribution. In other words, the model needs to "interpolate" and "extrapolate" at the same time. A higher value of $y$ provides a stronger signal that guides the diffusion model towards higher rewards, while the increasing distribution shift may hurt the generated samples' quality. In the sequel, we provide theoretical guarantees for the reward-conditioned diffusion models, aiming to answer the following question:

*How to provably estimate the reward-conditioned distribution via diffusion? How to balance the reward signal and distribution-shift effect, and ensure reward improvement in generated samples?*

**Our Approach.** To answer both questions, we consider a semi-supervised learning setting, where we are given a small dataset $\mathcal{D}_{\text{label}}$ with annotated rewards and a massive unlabeled dataset $\mathcal{D}_{\text{unlabel}}$. We estimate the reward function using $\mathcal{D}_{\text{label}}$ and then use the estimator for pseudo-labeling on $\mathcal{D}_{\text{unlabel}}$. Then we train a reward-conditioned diffusion model using the pseudo-labeled data. Our approach is illustrated in Figure 1. In real-world applications, there are other ways to incorporate the knowledge from the massive dataset $\mathcal{D}_{\text{unlabel}}$, e.g., finetuning from a pre-trained model [35, 54]. We focus on the pseudo-labeling approach, as it provides a cleaner formulation and exposes the error dependency on data size and distribution shift. The intuition behind and the message are applicable to other semi-supervised approaches; see experiments in Section 5.2.

From a theoretical standpoint, we consider data point $x$ having a latent linear representation. Specifically, we assume $x = Az$ for some matrix $A$ with orthonormal columns and $z$ being a latent variable. The latent variable often has a smaller dimension, reflecting the fact that practical data sets often exhibit intrinsic low-dimensional structures [13, 48, 37]. The representation matrix $A$ should be learned to promote sample efficiency and generation quality [8]. Our theoretical analysis reveals an intricate interplay between reward guidance, distribution shift, and implicit representation learning; see Figure 2 for illustration.

**Contributions.** Our results are summarized as follows.

**(1)** We show that the reward-conditioned diffusion model implicitly learns the latent subspace representation of $x$. Consequently, the model provably generates high-fidelity data that stay close to the subspace (Theorem 4.5).

**(2)** Given a target reward value, we analyze the statistical error of reward-directed generation, measured by the difference between the target value and the average reward of the generated population. In the case of a linear reward model, we show that this error includes the suboptimality gap of linear off-policy bandits with full knowledge of the subspace feature, if taking the target to be the maximum possible. In addition, the other two components of this error correspond to the distribution shift in score matching and the cost of off-support extrapolation (Theorem 4.6).

**(3)** We further extend our theory to nonparametric reward and distribution configurations where reward prediction and score matching are approximated by general function class, which covers the wildly adopted ReLU Neural Networks in real-world implementation (Section 4.3 and Appendix F).

**(4)** We provide numerical experiments under both synthesized setting and more realistic settings such as text-to-image generation (stable diffusion) and reinforcement learning (decision-diffuser) to support our theory (Section 5 and Appendix I).

To our best knowledge, our results present the first statistical theory for conditioned diffusion models and provably reward improvement guarantees for reward-directed generation.

## 2  Related Work

**Guided Diffusions.**   For image generations, guiding the backward diffusion process towards higher log probabilities predicted by a classifier (which can be viewed as the reward signal) leads to improved sample quality, where the classifier can either be separated trained, i.e., classifier-guided [11] or implicitly specified by the conditioned diffusion models, i.e., classifier-free [18]. Classifier-free guidance has become a standard technique in the state-of-the-art text-to-image diffusion models [39, 38, 4]. Other types of guidance are also explored [33, 14]. Similar ideas have been explored in

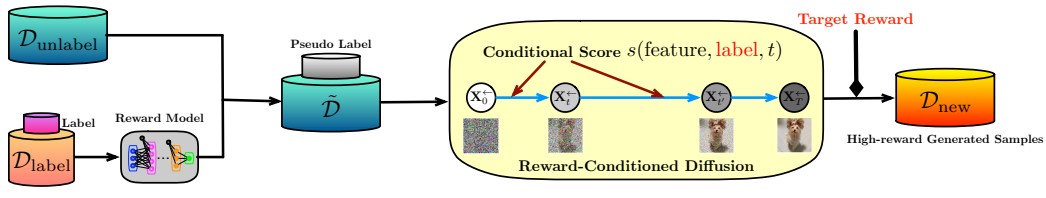

Step 1: Reward Learning     Step 2: Pseudo Labeling     Step 3: Conditional Diffusion Model Training     Step 4: Guided Generation

Figure 1: **Overview of reward-directed generation via conditional diffusion model.** We estimate the reward function from the labeled dataset. Then we compute the estimated reward for each instance of the unlabeled dataset. Finally, we train a reward-conditioned diffusion model using the pseudo-labeled data. Using the reward-conditioned diffusion model, we are able to generate high-reward samples.

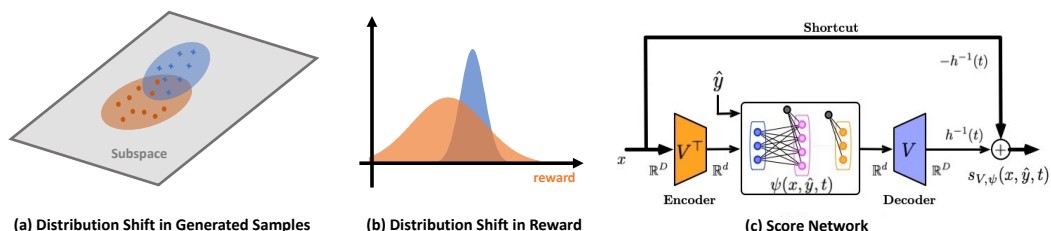

(a) Distribution Shift in Generated Samples     (b) Distribution Shift in Reward     (c) Score Network

Figure 2: **Illustrations of distribution shifts in samples, reward, and encoder-decoder score network.** When performing reward-directed conditional diffusion, **(a)** the distribution of the generated data shifts, but still stays close to the feasible data support; **(b)** the distribution of rewards for the next generation shifts and the mean reward improves. **(c)** (Adapted from [8]) the score network for reward-directed conditioned diffusion adopts an Encoder-Decoder structure.

sequence modelling problems. In offline reinforcement learning, Decision Diffuser [1] is a diffusion model trained on offline trajectories and can be conditioned to generate new trajectories with high returns, satisfying certain safety constraints, or composing skills. For discrete generations, Diffusion LM [27] manages to train diffusion models on discrete text space with an additional embedding layer and a rounding step. The authors further show that gradients of any classifier can be incorporated to control and guide the text generation.

**Theory of Diffusion Models** A line of work studies diffusion models from a sampling perspective. When assuming access to a score function that can accurately approximate the ground truth score function in $L^\infty$ or $L^2$ norm, [9, 25] provide polynomial convergence guarantees of score-based diffusion models. "Convergence of denoising diffusion models under the manifold hypothesis" by Valentin De Bortoli further studies diffusion models under the manifold hypothesis. Recently, [8] and [34] provide an end-to-end analysis of diffusion models. In particular, they develop score estimation and distribution estimation guarantees using the estimated score function. These results largely motivate our theory, whereas, we are the first to consider conditional score matching and statistical analysis of conditional diffusion models.

**Connection to Offline Bandit/RL** Our off-policy regret analysis of generated samples is related to offline bandit/RL theory [30, 29, 6, 12, 22, 32, 5]. In particular, our theory extensively deals with distribution shift in the offline data set by class restricted divergence measures, which are commonly adopted in offline RL. Moreover, our regret bound of generated samples consists of an error term that coincides with off-policy linear bandits. However, our analysis goes far beyond the scope of bandit/RL.

## 3 Reward-Directed Generation via Conditional Diffusion Models

In this section, we develop a conditioned diffusion model-based method to generate high-fidelity samples with desired properties. In real-world applications such as image/text generation and protein design, one often has access to abundant unlabeled data, but relatively limited number of labeled data. This motivates us to consider a semi-supervised learning setting.

**Notation**: $P_{xy}$ denotes ground truth joint distribution of $x$ and its label $y$, $P_x$ is the marginal of $x$. Any piece of data in $\mathcal{D}_{\text{label}}$ follows $P_{xy}$ and any data in $\mathcal{D}_{\text{unlabel}}$ follows $P_x$. $P$ is used to denote a distribution and $p$ denotes its corresponding density. $P(x \mid y = a)$ and $P(x, y = a)$ are the conditionals of $P_{xy}$ Similarly, we also use notation $P_{x\widehat{y}}$, $P(x \mid \widehat{y} = a)$ for the joint and conditional of $(x, \widehat{y})$, where $\widehat{y}$ is predicted by the learnt reward model. Also, denote a generated distribution using diffusion by $\widehat{P}$ (density $\widehat{p}$) followed by the same argument in parentheses as the true distribution it approximates, e.g. $\widehat{P}(x \mid y = a)$ is generated as an approximation of $P(x \mid y = a)$.

## 3.1 Problem Setup

Suppose we are given an unlabeled data set $\mathcal{D}_{\text{unlabel}} = \{x_j\}_{j=1}^{n_1}$ and a labeled data set $\mathcal{D}_{\text{label}} = \{(x_i, y_i)\}_{i=1}^{n_2}$, where it is often the case that $n_1 \gg n_2$. Assume without loss of generality that $\mathcal{D}_{\text{label}}$ and $\mathcal{D}_{\text{unlabel}}$ are independent. In both datasets, suppose $x$ is sampled from an unknown population distribution $P_x$. In our subsequent analysis, we focus on the case where $P_x$ is supported on a latent subspace, meaning that the raw data $x$ admits a low-dimensional representation (see Assumption 4.1). We model $y$ as a noisy measurement of a reward function determined by $x$, given by

$$y = f^*(x) + \epsilon \quad \text{for} \quad \epsilon \sim \mathsf{N}(0, \sigma^2) \quad \text{with} \quad 1 > \sigma > 0.$$

A user can specify a target reward value, i.e., $y = a$. Then the objective of directed generation is to sample from the conditional distribution $P(x|y = a)$. Given $f^*, P_x$ or the low-dimensional support of $P_x$ are unknown, we need to learn these unknowns explicitly and implicitly through reward-conditioned diffusion.

## 3.2 Meta Algorithm

---

**Algorithm 1** Reward-Conditioned Generation via Diffusion Model (RCGDM)

---

1: **Input**: Datasets $\mathcal{D}_{\text{unlabel}}, \mathcal{D}_{\text{label}}$, target reward value $a$, early-stopping time $t_0$, noise level $\nu$. (Note: in the following psuedo-code, $\phi_t(x)$ is the Gaussian density and $\eta$ is the step size of discrete backward SDE, see §3.3 for elaborations on conditional diffusion)

2: **Reward Learning**: Estimate the reward function by

$$\widehat{f} \in \underset{f \in \mathcal{F}}{\operatorname{argmin}} \sum_{(x_i, y_i) \in \mathcal{D}_{\text{label}}} \ell(f(x_i), y_i), \tag{3.1}$$

where $\ell$ is a loss and $\mathcal{F}$ is a function class.

3: **Pseudo labeling**: Use the learned function $\widehat{f}$ to evaluate unlabeled data $\mathcal{D}_{\text{unlabel}}$ and augment it with pseudo labeles: $\widetilde{\mathcal{D}} = \{(x_j, \widehat{y}_j) = \widehat{f}(x_j) + \xi_j\}_{j=1}^{n_1}$ for $\xi_j \overset{\text{i.i.d.}}{\sim} \mathsf{N}(0, \nu^2)$.

4: **Conditional score matching**: Minimize over $s \in \mathcal{S}$ ($\mathcal{S}$ constructed as 3.8) on data set $\widetilde{\mathcal{D}}$ via

$$\widehat{s} \in \underset{s \in \mathcal{S}}{\operatorname{argmin}} \int_{t_0}^{T} \widehat{\mathbb{E}}_{(x,\widehat{y}) \in \widetilde{\mathcal{D}}} \mathbb{E}_{x' \sim \mathsf{N}(\alpha(t)x, h(t)I_D)} \left[ \|\nabla_{x'} \log \phi_t(x'|x) - s(x', \widehat{y}, t)\|_2^2 \right] \mathrm{d}t. \tag{3.2}$$

5: **Conditioned generation**: Use the estimated score $\widehat{s}(\cdot, a, \cdot)$ to sample from the backward SDE:

$$\mathrm{d}\widetilde{X}_t^{t,\Leftarrow} = \left[ \frac{1}{2} \widetilde{X}_{k\eta}^{y,\Leftarrow} + \widehat{s}(\widetilde{X}_{k\eta}^{y,\Leftarrow}, a, T - k\eta) \right] \mathrm{d}t + \mathrm{d}\overline{W}_t \quad \text{for } t \in [k\eta, (k+1)\eta], k \in \lfloor \frac{T}{\eta} \rfloor. \tag{3.3}$$

6: **Return**: Generated population $\widehat{P}(\cdot|\widehat{y} = a)$, learned subspace representation $V$ contained in $\widehat{s}$.

---

In order to generate novel samples with both high fidelity and high rewards, we propose Reward-Conditioned Generation via Diffusion Models (RCGDM); see Algorithm 1 for details. By using the labeled data $\mathcal{D}_{\text{label}}$, we approximately estimate the reward function $f^*$ by regression, then we obtain an estimated reward function $\widehat{f}$. We then use $\widehat{f}$ to augment the unlabeled data $\mathcal{D}_{\text{unlabel}}$ with "pseudo labeling" and additive noise, i.e., $\widetilde{\mathcal{D}} = \{(x_j, \widehat{y}_j = \widehat{f}(x_j) + \xi_j)\}_{j=1}^{n_1}$ with $\xi_j \sim \mathsf{N}(0, \nu^2)$ of a small variance $\nu^2$. Here, we added noise $\xi_j$ merely for technical reasons in the proof. We denote the

joint distribution of $(x, \widehat{y})$ as $P_{x\widehat{y}}$. Next, we train a conditional diffusion model using the augmented dataset $\widetilde{D}$. If we specify a target value of the reward, for example letting $y = a$, we can generate conditioned samples from the distribution $\widehat{P}(x|\widehat{y} = a)$ by backward diffusion.

**Alternative approaches.** In Line 4, Algorithm 1 trains the conditional diffusion model via conditional score matching. This approach is suitable when we have access to the unlabeled dataset and need to train a brand-new diffusion model from scratch. Empirically, in order to realize conditional generation, we can utilize a pre-trained diffusion model and incorporate with reward signals to be conditioned on. Existing methods falling in to this category include classifier-based guidance [11], fine-tuning [54], and self-distillation [47]. For theoretical cleanness, we focus on analysing Algorithm 1 as it shares the same core essence with other alternative methods, which is approximating of the conditional score $\nabla \log p_t(x_t|y)$.

## 3.3 Training of Conditional Diffusion Model

In this section, we provide details about the training and sampling of conditioned diffusion in Algorithm 1 (Line 4: conditional score matching and Line 5: conditional generation). In Algorithm 1, conditional diffusion model is learned with $\widetilde{\mathcal{D}} = \{(x_j, \widehat{y}_j = \widehat{f}(x_j) + \xi_j)\}_{j=1}^{n_1}$, where $(x, \widehat{y}) \sim P_{x\widehat{y}}$. For simplicity, till the end of this section we use $y$ instead of $\widehat{y}$ to denote the condition variable. The diffusion model is to approximate the conditional probability $P(x \mid \widehat{y})$.

**Conditional Score Matching.** The working flow of conditional diffusion models is nearly identical to that of unconditioned diffusion models reviewed in Appendix A. A major difference is we learn a conditional score $\nabla \log p_t(x|y)$ instead of the unconditional one. Here $p_t$ denotes the marginal density function at time $t$ of the following forward O-U process,

$$\mathrm{d}X_t^y = -\frac{1}{2}g(t)X_t^y \mathrm{d}t + \sqrt{g(t)}\mathrm{d}W_t \quad \text{with} \quad X_0^y \sim P_0(x|y) \text{ and } t \in (0, T], \qquad (3.4)$$

where similarly $T$ is a terminal time, $(W_t)_{t\geq 0}$ is a Wiener process, and the initial distribution $P_0(x|y)$ is induced by the $(x, \widehat{y})$-pair distribution $\bar{P}_{x\widehat{y}}$. Note here the noise is only added on $x$ but not on $y$. Throughout the paper, we consider $g(t) = 1$ for simplicity. We denote by $P_t(x_t|y)$ the distribution of $X_t^y$ and let $p_t(x_t|y)$ be its density and $P_t(x_t, y)$ be the corresponding joint, shorthanded as $P_t$. A key step is to estimate the unknown $\nabla \log p_t(x_t|y)$ through denoising score matching [46]. A conceptual way is to minimize the following quadratic loss with $\mathcal{S}$, a concept class.

$$\underset{s \in \mathcal{S}}{\operatorname{argmin}} \int_0^T \mathbb{E}_{(x_t, y) \sim P_t} \left[ \|\nabla \log p_t(x_t|y) - s(x_t, y, t)\|_2^2 \right] \mathrm{d}t, \qquad (3.5)$$

Unfortunately, the loss in (3.5) is intractable since $\nabla \log p_t(x_t|y)$ is unknown. Inspired by Hyvärinen and Dayan [20] and Vincent [52], we choose a new objective (3.2) and show their equivalence in the following Proposition. The proof is provided in Appendix C.1.

**Proposition 3.1 (Score Matching Objective for Implementation).** For any $t > 0$ and score estimator $s$, there exists a constant $C_t$ independent of $s$ such that

$$\mathbb{E}_{(x_t, y) \sim P_t} \left[ \|\nabla \log p_t(x_t|y) - s(x_t, y, t)\|_2^2 \right]$$
$$= \mathbb{E}_{(x, y) \sim P_{x\widehat{y}}} \mathbb{E}_{x' \sim \mathsf{N}(\alpha(t)x, h(t)I_D)} \left[ \|\nabla_{x'} \log \phi_t(x'|x) - s(x', y, t)\|_2^2 \right] + C_t, \qquad (3.6)$$

where $\nabla_{x'} \log \phi_t(x'|x) = -\frac{x' - \alpha(t)x}{h(t)}$, where $\phi_t(x'|x)$ is the density of $\mathsf{N}(\alpha(t)x, h(t)I_D)$ with $\alpha(t) = \exp(-t/2)$ and $h(t) = 1 - \exp(-t)$.

Equation (3.6) allows an efficient implementation, since $P_{x\widehat{y}}$ can be approximated by the empirical data distribution in $\widetilde{\mathcal{D}}$ and $x'$ is easy to sample. Integrating (3.6) over time $t$ leads to a practical conditional score matching object

$$\underset{s \in \mathcal{S}}{\operatorname{argmin}} \int_{t_0}^T \widehat{\mathbb{E}}_{(x, y) \sim P_{x\widehat{y}}} \mathbb{E}_{x' \sim \mathsf{N}(\alpha(t)x, h(t)I_D)} \left[ \|\nabla_{x'} \log \phi_t(x'|x) - s(x', y, t)\|_2^2 \right] \mathrm{d}t, \qquad (3.7)$$

where $t_0 > 0$ is an early-stopping time to stabilize the training [44, 50] and $\widehat{\mathbb{E}}$ denotes the empirical distribution.

Constructing a function class $\mathcal{S}$ adaptive to data structure is beneficial for learning the conditional score. In the same spirit of [8], we propose the score network architecture (see Figure 2(c) for an illustration):

$$\mathcal{S} = \left\{ \mathbf{s}_{V,\psi}(x,y,t) = \frac{1}{h(t)}(V \cdot \psi(V^\top x, y, t) - x) : V \in \mathbb{R}^{D \times d}, \ \psi \in \Psi : \mathbb{R}^{d+1} \times [t_0, T] \to \mathbb{R}^d \right\}, \quad (3.8)$$

with $V$ being any $D \times d$ matirx with orthonormal columns and $\Phi$ a customizable function class. This design has a linear encoder-decoder structure, catering for the latent subspace structure in data. Also $-\frac{1}{h(t)}x$ is includes as a shortcut connection.

**Conditioned Generation.** Sampling from the model is realized by running a discretized backward process with step size $\eta > 0$ described as follows:

$$d\widetilde{X}_t^{t,\Leftarrow} = \left[ \frac{1}{2}\widetilde{X}_{k\eta}^{y,\Leftarrow} + \widehat{s}(\widetilde{X}_{k\eta}^{y,\Leftarrow}, y, T - k\eta) \right] dt + d\overline{W}_t \quad \text{for} \quad t \in [k\eta, (k+1)\eta]. \quad (3.3 \text{ revisited})$$

initialized with $\widetilde{X}_t^{t,\Leftarrow} \sim \mathsf{N}(0, I_D)$ and $\overline{W}_t$ is a reversed Wiener process. Note that in (3.3), the unknown conditional score $\nabla p_t(x|y)$ is substituted by $\widehat{s}(x, y, t)$.

# 4 Main Theory

In this section, we analyze the conditional generation process specified by Algorithm 1. We will focus on the scenario where samples $x$ admit a low-dimensional subspace representation, stated as the following assumption.

**Assumption 4.1** . Data sampling distribution $P_x$ is supported on a low-dimensional linear subspace, i.e., $x = Az$ for an unknown $A \in \mathbb{R}^{D \times d}$ with orthonormal columns and $z \in \mathbb{R}^d$ is a latent variable.

Note that our setup covers the full-dimensional setting as a special case when $d = D$. Yet the case of $d < D$ is much more interesting, as practical datasets are rich in intrinsic geometric structures [13, 37, 48]. Furthermore, the representation matrix $A$ may encode critical constraints on the generated data. For example, in protein design, the generated samples need to be similar to natural proteins and abide rules of biology, otherwise they easily fail to stay stable, leading to substantial reward decay. In those applications, off-support data may be risky and suffer from a large degradation of rewards, which we model using a function $h$ as follows.

**Assumption 4.2** . The ground truth reward $f^*(x) = g^*(x_\parallel) - h^*(x_\perp)$, where $g^*(x_\parallel) = (\theta^*)^\top x_\parallel$ where $\theta^* = A\beta^*$ for some $\beta^* \in \mathbb{R}^d$ and $\|\theta^*\|_2 = \|\beta^*\|_2 = 1$ and $h^*(x_\perp)$ is non-decreasing in terms of $\|x_\perp\|_2$ with $h^*(0) = 0$.

Assumption 4.2 adopts a simple linear reward model for ease of presentation. In this case, we estimate $\theta^*$ by ridge regression, and (3.1) in Algorithm 1 becomes $\widehat{\theta} = \arg\min_\theta \sum_{i=1}^{n_2}(\theta^\top x_i - y_i)^2 + \lambda\|\theta\|_2^2$ for a positive coefficient $\lambda$. Later in Section 3.3 and Appendix F, we extend our results beyond linear models to deep ReLU networks.

## 4.1 Conditional DM Learns Subspace Representation

Recall that Algorithm 1 has two outputs: generated population $\widehat{P}(\cdot|\widehat{y} = a)$ and learned representation matrix $V$. Use notation $\widehat{P}_a := \widehat{P}(\cdot|\widehat{y} = a)$ (generated distribution) and $P_a := P(\cdot|\widehat{y} = a)$ (target distribution) for better clarity in result presentation. To assess the quality of subspace learning, we utilize two metrics defined as

$$\angle(V, A) = \|VV^\top - AA^\top\|_F^2 \quad \text{and} \quad \mathbb{E}_{x \sim \widehat{P}_a}[\|x_\perp\|_2]. \quad (4.1)$$

$\angle(V, A)$ is defined for matrices $V, A$, where $A$ is the matrix encoding the ground truth subspace. Clearly, $\angle(V, A)$ measures the difference in the column span of $V$ and $A$, which is also known as the subspace angle. $\mathbb{E}_{x \sim \widehat{P}_a}[\|x_\perp\|]$ is defined as the expected $l_2$ distance between $x$ and the true subspace. Theorem 4.5 provides guarantees on this two metrics under following assumptions, proof and Interpretation of Theorem 4.5 are deferred to Appendix D.2.

To ease the presentation, we consider a Gaussian design on $x$, i.e. the latent $z$ is Gaussian as stated in Assumption 4.4. Since our guarantee on $\angle(V, A)$ holds under milder assumption than Gaussian, we also list the Assumption 4.3.

**Assumption 4.3** . The latent variable $z$ follows distribution $P_z$ with density $p_z$, such that there exists constants $B, C_1, C_2$ verifying $p_z(z) \leq (2\pi)^{-(d+1)/2} C_1 \exp\left(-C_2 \|z\|_2^2/2\right)$ whenever $\|z\|_2 > B$. And ground truth score is realizable: $\nabla \log p_t(x \mid \widehat{y}) \in \mathcal{S}$.

**Assumption 4.4** . Further assume $z \sim \mathsf{N}(0, \Sigma)$ with its covariance matrix $\Sigma$ satisfying $\lambda_{\min} I_d \preceq \Sigma \preceq \lambda_{\max} I_d$ for $0 < \lambda_{\min} \leq \lambda_{\max} \leq 1$.

**Theorem 4.5** (**Subspace Fidelity of Generated Data**). Under Assumption 4.1, if Assumption 4.3 holds with $c_0 I_d \preceq \mathbb{E}_{z \sim P_z}\left[zz^\top\right]$, then with high probability on data,

$$\angle(V, A) = \widetilde{\mathcal{O}}\left(\frac{1}{c_0} \sqrt{\frac{\mathcal{N}(\mathcal{S}, 1/n_1) D}{n_1}}\right) \tag{4.2}$$

with $\mathcal{N}(\mathcal{S}, 1/n_1)$ being the log covering number of function class $\mathcal{S}$ as in (3.8). When Assumption 4.4 holds, $\mathcal{N}(\mathcal{S}, 1/n_1) = \mathcal{O}((d^2 + Dd)\log(Ddn_1))$ and thus $\angle(V, A) = \widetilde{\mathcal{O}}(\frac{1}{\lambda_{\min}} \sqrt{\frac{(Dd^2 + D^2 d)}{n_1}})$. Further under Assumption 4.2, it holds that

$$\mathbb{E}_{x \sim \widehat{P}_a}[\|x_\perp\|_2] = \mathcal{O}\left(\sqrt{t_0 D} + \sqrt{\angle(V, A)} \cdot \sqrt{\frac{a^2}{\|\beta^*\|_\Sigma} + d}\right), \tag{4.3}$$

where $\beta^*$ is groundtruth parameter of linear model.

## 4.2 Provable Reward Improvement via Conditional Generation

Let $y^*$ be a target reward value and $P$ be a generated distribution. Define the suboptimality of $P$ as

$$\mathtt{SubOpt}(P; y^*) = y^* - \mathbb{E}_{x \sim P}[f^*(x)],$$

which measures the gap between the expected reward of $x \sim P$ and the target value $y^*$. In the language of bandit learning, this gap can also be viewed as a form of *off-policy regret*. Given a target value $y^* = a$, we want to derive guarantees for $\mathtt{SubOpt}(\widehat{P}_a; y^* = a)$, recall $\widehat{P}_a := \widehat{P}(\cdot|\widehat{y} = a)$ denotes the generated distribution. In Theorem 4.6, we show $\mathtt{SubOpt}(\widehat{P}_a; y^* = a)$ comprises of three components: off-policy bandit regret which comes from the estimation error of $\widehat{f}$, on-support and off-support errors coming from approximating conditional distributions with diffusion.

**Theorem 4.6** (**Off-policy Regret of Generated Samples**). Suppose Assumption 4.1, 4.2 and 4.4 hold. We choose $\lambda = 1$, $t_0 = \left((Dd^2 + D^2 d)/n_1\right)^{1/6}$ and $\nu = 1/\sqrt{D}$. With high probability, running Algorithm 1 with a target reward value $a$ gives rise to

$$\mathtt{SubOpt}(\widehat{P}_a; y^* = a)$$

$$\leq \underbrace{\sqrt{\mathrm{Tr}(\widehat{\Sigma}_\lambda^{-1} \Sigma_{P_a})} \cdot \mathcal{O}\left(\sqrt{\frac{d \log n_2}{n_2}}\right)}_{\mathcal{E}_1 : \text{off-policy bandit regret}} + \underbrace{\left|\mathbb{E}_{P_a}[g^*(x_\|)] - \mathbb{E}_{\widehat{P}_a}[g^*(x_\|)]\right|}_{\mathcal{E}_2 : \text{on-support diffusion error}} + \underbrace{\mathbb{E}_{\widehat{P}_a}[h^*(x_\perp)]}_{\mathcal{E}_3 : \text{off-support diffusion error}},$$

$$\tag{4.4}$$

where $\widehat{\Sigma}_\lambda := \frac{1}{n_2}(X^\top X + \lambda I)$ where $X$ is the stack matrix of $\mathcal{D}_{\text{label}}$ and $\Sigma_{P_a} = \mathbb{E}_{P_a}[xx^\top]$.

**Implications and Discussions:** (1) Equation (4.4) decomposes the suboptimality gap into two separate parts of error: error from reward learning ($\mathcal{E}_1$) and error coming from diffusion ($\mathcal{E}_2$ and $\mathcal{E}_3$).

(2) It is also worth mentioning that $\mathcal{E}_2$ and $\mathcal{E}_3$ depend on $t_0$ and that by taking $t_0 = \left((Dd^2 + D^2 d)/n_1\right)^{1/6}$ one gets a good trade-off and small $\mathcal{E}_2 + \mathcal{E}_3$.

(3) $\mathcal{E}_1$ suggests diffusion model is essentially doing representation learning, reducing $D$ to smaller latent dimension $d$. It can be seen from $\mathrm{Tr}(\widehat{\Sigma}_\lambda^{-1} \Sigma_{p_q}) \leq \mathcal{O}\left(\frac{a^2}{\|\beta^*\|_\Sigma} + d\right)$ when $n_2 = \Omega(\frac{1}{\lambda_{min}})$.

(3) If we ignore the diffusion errors when $n_1$ is large enough, the suboptimatliy gap resembles the suboptimatliy of off-policy bandit learning in feature subspace [22, Section 3.2], [32, 5]. It shows

the major source of error occurs when moving towards the target distributions and the error behaves similar to bandit.

**(4)** On-support diffusion error entangles with distribution shift in complicated ways. We show

$$\mathcal{E}_2 = \left( \mathtt{DistroShift}(a) \cdot \left( d^2 D + D^2 d \right)^{1/6} n_1^{-1/6} \cdot a \right),$$

where $\mathtt{DistroShift}(a)$ quantifies the distribution shift depending on different reward values. In the special case of the latent covariance matrix $\Sigma$ is known, we can quantify the distribution shift as $\mathtt{DistroShift}(a) = \mathcal{O}(a \vee d)$. We observe an interesting phase shift. When $a < d$, the training data have a sufficient coverage with respect to the generated distribution $\widehat{P}_a$. Therefore, the on-support diffusion error has a lenient linear dependence on $a$. However, when $a > d$, the data coverage is very poor and $\mathcal{E}_2$ becomes quadratic in $a$, which quickly amplifies.

**(5)** When generated samples deviate away from the latent space, the reward may substantially degrade as determined by the nature of $h$.

To the authors' best knowledge, this is a first theoretical attempt to understand reward improvement of conditional diffusion. These results imply a potential connection between diffusion theory and off-policy bandit learning, which is interesting for more future research. See proofs in Appendix D.3.

### 4.3 Extension to Nonparametric Function Class

Our theoretical analysis, in its full generality, extends to using general nonparametric function approximation for both the reward and score functions. To keep our paper succinct, we refer to Appendix F and Theorem F.4 for details of our nonparametric theory for reward-conditioned generation. Informally, the regret of generated samples is bounded by

$$\mathtt{SubOpt}(\widehat{P}_a; y^* = a) = \widetilde{\mathcal{O}} \left( \mathtt{DistroShift}(a) \cdot \left( n_2^{-\frac{\alpha}{2\alpha+d}} + n_1^{-\frac{2}{3(d+6)}} \right) \right) + \mathbb{E}_{\widehat{P}_a}[h^*(x_\perp)]$$

with high probability. Additionally, the nonparamtric generators is able to estimate the representation matrix $A$ up to an error of $\angle(V, A) = \widetilde{\mathcal{O}}(n_1^{-\frac{2}{d+6}})$. Here the score is assumed to be Lipschitz continuous and $\alpha$ is the smoothness parameter of the reward function, and $\mathtt{DistroShift}(a)$ is a class-restricted distribution shift measure. Our results on nonparametric function approximation covers the use of deep ReLU networks as special cases.

## 5 Numerical Experiments

### 5.1 Simulation

We first perform the numerical simulation of Algorithm 1 following the setup in Assumption 4.1, 4.2 and 4.4. We choose $d = 16, D = 64, g^\star(x) := 5\|x\|_2^2$, and generate $\beta^\star$ by uniformly sampling from the unit sphere. The latent variable $z$ is generated from $\mathsf{N}(0, \mathsf{I_d})$, which is then used to construct $x = Az$ with some randomly generated orthonormal matrix $A$. We use the 1-dimensional version of the UNet [40] to approximate the score function. More details are deferred to Appendix I.

Figure 3 shows the average reward of the generated samples under different target reward values. We also plot the distribution shift and off-support deviation in terms of the 2-norm distance from the support. For small target reward values, the generation average reward almost scales linearly with the target value, which is consistent with the theory as the distribution shift remains small for these target values. The generation reward begins to decrease as we further increase the target reward value, and the reason is two fold. Firstly, the off-support deviation of the generated samples becomes large in this case, which prevents the generation reward from further going up. Secondly, the distribution shift increases rapidly as we further increase the target value, making the theoretical guarantee no longer valid. In Figure 4, we show the distribution of the rewards in the generated samples. As we increase the target reward values, the generation rewards become less concentrated and are shifted to the left of the target value, which is also due to the distribution shift and off-support deviation.

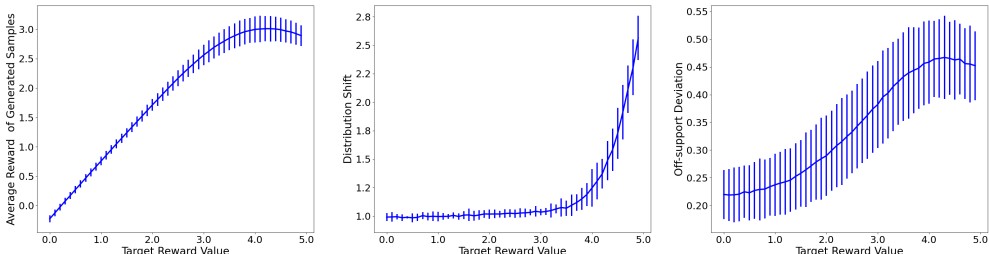

Figure 3: **Quality of generated samples as target reward value increases.** Left: Average reward of the generation; Middle: Distribution shift; Right: Off-support deviation. The errorbar is computed by 2 times the standard deviation over 5 runs.

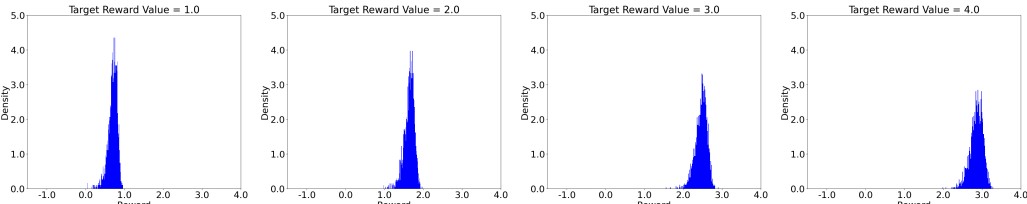

Figure 4: **Shifting reward distribution of the generated population.**

## 5.2 Directed Text-to-Image Generation

Next, we empirically verify our theory through directed text-to-image generation. Instead of training a diffusion model from scratch, we use Stable Diffusion v1.5 [39], pre-trained on LAION dataset [42]. Stable Diffusion operates on the latent space of its Variational Auto-Encoder and can incorporate text conditions. We show that by training a reward model we can further guide the Stable Diffusion model to generate images of desired properties.

**Ground-truth Reward Model.** We start from an ImageNet [10] pre-trained ResNet-18 [17] model and replace the final prediction layer with a randomly initialized linear layer of scalar outputs. Then we use this model as the ground-truth reward model. To investigate the meaning of this randomly-generated reward model, we generate random samples and manually inspect the images with high rewards and low rewards. The ground-truth reward model seems to favor colorful and vivid natural scenes against monochrome and dull images; see Appendix I for sample images.

**Labelled Dataset.** We use the ground-truth reward model to compute a scalar output for each instance in the CIFAR-10 [24] training dataset and perturb the output by adding a Gaussian noise from $\mathcal{N}(0, 0.01)$. We use the images and the corresponding outputs as the training dataset.

**Reward-network Training.** To avoid adding additional input to the diffusion model and tuning the new parameters, we introduce a new network $\mu_\theta$ and approximate $p_t(y|x_t)$ by $\mathsf{N}(\mu_\theta(\mathsf{x_t}), \sigma^2)$. For simplicity, we set $\sigma^2$ as a tunable hyperparameter. We share network parameters for different noise levels $t$, so our $\mu_\theta$ has no additional input of $t$. We train $\mu_\theta$ by minimizing the expected KL divergence between $p_t(y|x_t)$ and $\mathsf{N}(\mu_\theta(\mathsf{x_t}), \sigma^2)$:

$$\mathbb{E}_t \mathbb{E}_{x_t} \Big[ \mathrm{KL}(p_t(y|x_t) \mid \mathsf{N}(\mu_\theta(\mathsf{x_t}), \sigma^2)) \Big] = \mathbb{E}_t \mathbb{E}_{(\mathsf{x_t}, \mathsf{y}) \sim \mathsf{p_t}} \frac{\|\mathsf{y} - \mu_\theta(\mathsf{x_t})\|_2^2}{2\sigma^2} + \text{Constant}.$$

Equivalently, we train the reward model $\mu_\theta$ to predict the noisy reward $y$ from the noisy inputs $x_t$. Also, notice that the minimizers of the objective do not depend on the choice of $\sigma^2$.

**Reward-network-based Directed Diffusion.** To perform reward-directed conditional diffusion, observe that $\nabla_x \log p_t(x|y) = \nabla_x \log p_t(x) + \nabla_x \log p_t(y|x)$, and $p_t(y|x) \propto \exp\Big( -\frac{\|y - \mu_\theta(x)\|_2^2}{2\sigma^2} \Big)$. Therefore,

$$\nabla_x \log p_t(y|x) = -1/\sigma^2 \cdot \nabla_x \Big[ \frac{1}{2} \|y - \mu_\theta(x)\|_2^2 \Big].$$

In our implementation, we compute the gradient by back-propagation through $\mu_\theta$ and incorporate this gradient guidance into each denoising step of the DDIM sampler [43] following [11] (equation (14)). We see that $1/\sigma^2$ corresponds to the weights of the gradient with respect to unconditioned score. In the sequel, we refer to $1/\sigma^2$ as the "guidance level", and $y$ as the "target value".

**Quantitative Results.** We vary $1/\sigma^2$ in $\{25, 50, 100, 200, 400\}$ and $y$ in $\{1, 2, 4, 8, 16\}$. For each combination, we generate 100 images with the text prompt "A nice photo" and calculate the mean and the standard variation of the predicted rewards and the ground-truth rewards. The results are plotted in Figure 5. From the plot, we see similar effects of increasing the target value $y$ at different guidance levels $1/\sigma^2$. A larger target value puts more weight on the guidance signals $\nabla_x \mu_\theta(x)$, which successfully drives the generated images towards higher predicted rewards, but suffers more from the distribution-shift effects between the training distribution and the reward-conditioned distribution, which renders larger gaps between the predicted rewards and the ground-truth rewards. To optimally choose a target value, we must trade off between the two counteractive effects.

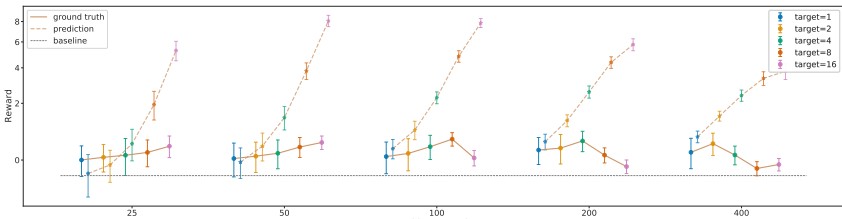

Figure 5: **The predicted rewards and the ground-truth rewards of the generated images**. At each guidance level, increasing the target $y$ successfully directs the generation towards higher predicted rewards, but also increases the error induced by the distribution shift. The reported baseline is the expected ground-truth reward for undirected generations.

**Qualitative Results.** To qualitatively test the effects of the reward conditioning, we generate a set of images with increasing target values $y$ under different text prompts and investigate the visual properties of the produced images. We isolate the effect of reward conditioning by fixing all the randomness during the generation processes, so the generated images have similar semantic layouts. After hyper-parameter tuning, we find that setting $1/\sigma^2 = 100$ and $y \in \{2, 4, 6, 8, 10\}$ achieves good results across different text prompts and random seeds. We pick out typical examples and summarized the results in Figure 6, which demonstrates that as we increase the target value, the generated images become more colorful at the expense of degradations of the image qualities.

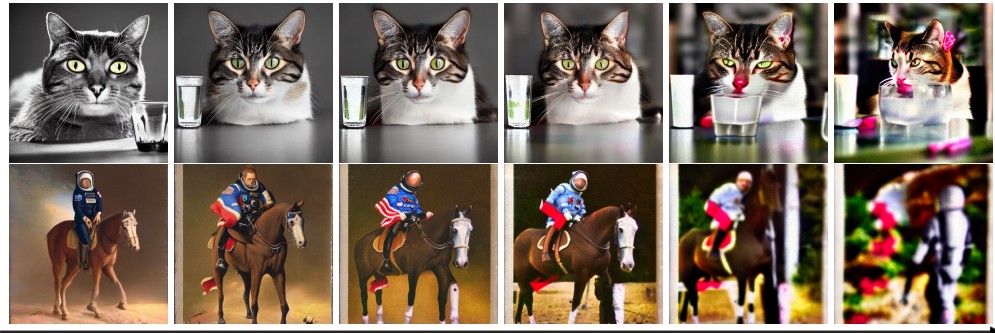

Figure 6: **The effects of the reward-directed diffusion.** Increasing the target value directs the images to be more colorful and vivid at the cost of degradation of the image qualities. Leftmost: without reward conditioning. Second-to-Last: target value $y = 2, 4, 6, 8, 10$. The guidance level $1/\sigma^2$ is fixed to 100. The text prompts are "A cat with a glass of water.", "An astronaut on the horseback".

## 6  Conclusion

In the paper, we study the problem of generating high-reward and high-quality samples using reward-directed conditional diffusion models, focusing on the semi-supervised setting where massive unlabeled data and limited labeled data are given. We provide theoretical results for subspace recovery and reward improvement, demonstrating the trade-off between the strength of the reward target and the distribution shift. Numerical results support our theory well.

## Acknowledgments and Disclosure of Funding

Mengdi Wang acknowledges the support by NSF grants CPS-2312093, DMS-1953686, IIS-2107304, CMMI1653435, ONR grant 1006977 and C3.AI.

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

**Index of Supplementary Materials**

# A    Preliminaries

## A.1    Diffusion Models

We first provide a brief review of diffusion models and its training/sampling procedure. We consider diffusion in continuous time [23, 46], where diffusion is described as forward and backward SDEs.

**Forward SDE and Score Matching.** In the forward process, noise is added to original data progressively as an Ornstein-Ulhenbeck process for instance:

$$\mathrm{d}X_t = -\frac{1}{2}g(t)X_t\mathrm{d}t + \sqrt{g(t)}\mathrm{d}W_t \ \ \text{for} \ \ g(t) > 0, \tag{A.1}$$

where initial $X_0 \sim P_{\text{data}}$ and $(W_t)_{t\geq0}$ is a standard Wiener process, and $g(t)$ is a nondecreasing weighting function. In practice, the forward process (A.1) terminates at a sufficiently large $T > 0$ such that the corrupted $X_T$ is close to the standard Gaussian $\mathsf{N}(\mathbf{0}, I_D)$. To enable data generation in future, the score $\nabla \log p_t(\cdot)$ at $t$ is the key to learn, here $p_t$ denotes the marginal density of $X_t$. We often use an estimated score function $\widehat{s}(\cdot, t)$ trained by minimizing a score matching loss.

**Backward SDE for Generation.** Diffusion models generate samples through a backward SDE (A.2) reversing the time in (A.1) [2, 16], i.e.,

$$\mathrm{d}X_t^{\leftarrow} = \left[\frac{1}{2}g(T-t)X_t^{\leftarrow} + g(T-t)\nabla \log p_{T-t}(X_t^{\leftarrow})\right]\mathrm{d}t + \sqrt{g(T-t)}\mathrm{d}\overline{W}_t, \tag{A.2}$$

where $\overline{W}_t$ is a reversed Wiener process. In practice, the backward process is initialized with $\mathsf{N}(0, I_D)$ and the unknown conditional score $\nabla \log p_t(\cdot)$ is replaced by an estimated counterpart $\widehat{s}(\cdot, t)$.

# B    Limitations

We do not see outstanding limitations in our analysis. The linear subspace assumption initiates the study of conditional diffusion models on low-dimensional data. We expect to stimulate more sophisticated analyses under general assumptions such as manifold data.

# C    Omitted Proof in Section 3

## C.1    Proof of Proposition 3.1

*Proof.* For any $t \geq 0$, it hold that $\nabla_{x_t} \log p_t(x_t \mid y) = \nabla_{x_t} \log p_t(x_t, y)$ since the gradient is taken w.r.t. $x_t$ only. Then plugging in this equation and expanding the norm square on the LHS gives

$$\mathbb{E}_{(x_t,y)\sim P_t}\left[\|\nabla_{x_t} \log p_t(x_t, y) - s(x_t, y, t)\|_2^2\right] = \mathbb{E}_{(x_t,y)\sim P_t}\Big[\|s(x_t, y, t)\|_2^2$$
$$- 2\langle\nabla_{x_t} \log p_t(x_t, y), s(x_t, y, t)\rangle\Big] + C.$$

Then it suffices to prove

$$\mathbb{E}_{(x_t,y)\sim P_t}\left[\langle \nabla_{x_t}\log p_t(x_t,y), s(x_t,y,t)\rangle\right] = \mathbb{E}_{(x,y)\sim P_{x\widehat{y}}}\mathbb{E}_{x'\sim \mathsf{N}(\alpha(t)x,h(t)I)}\left[\langle \nabla_{x'}\phi_t(x'\mid x), s(x',y,t)\rangle\right]$$

Using integration by parts to rewrite the inner product we have

$$\mathbb{E}_{(x_t,y)\sim P_t}\left[\langle \nabla_{x_t}\log p_t(x_t,y), s(x_t,y,t)\rangle\right] = \int p_t(x_t,y)\langle \nabla_{x_t}\log p_t(x_t,y), s(x_t,y,t)\rangle dx_t dy$$

$$= \int \langle \nabla_{x_t} p_t(x_t,y), s(x_t,y,t)\rangle dx_t dy$$

$$= -\int p_t(x_t,y)\,\mathrm{div}(s(x_t,y,t))dx_t dy,$$

where denote by $\phi_t(x'|x)$ the density of $\mathsf{N}(\alpha(t)x,h(t)I_D)$ with $\alpha(t)=\exp(-t/2)$ and $h(t)=1-\exp(-t)$, then

$$-\int p_t(x_t,y)\,\mathrm{div}(s(x_t,y,t))dx_t dy = -\mathbb{E}_{(x,y)\sim P_{x\widehat{y}}}\int \phi_t(x'\mid x)\,\mathrm{div}(s(x',y,t))dx'$$

$$= \mathbb{E}_{(x,y)\sim P_{x\widehat{y}}}\int \langle \nabla_{x'}\phi_t(x'\mid x), s(x',y,t)\rangle dx'$$

$$= \mathbb{E}_{(x,y)\sim P_{x\widehat{y}}}\mathbb{E}_{x'\sim \mathsf{N}(\alpha(t)x,h(t)I)}\left[\langle \nabla_{x'}\phi_t(x'\mid x), s(x',y,t)\rangle\right].$$

$\square$

## D   Omitted Proofs in Section 4

**Additional Notations:**   We follow the notations in the main paper along with some additional ones. Use $P_t^{LD}(z)$ to denote the low-dimensional distribution on $z$ corrupted by diffusion noise. Formally, $p_t^{LD}(z) = \int \phi_t(z'|z)p_z(z)dz$ with $\phi_t(\cdot|z)$ being the density of $\mathsf{N}(\alpha(t)z,h(t)I_d)$. $P_{t_0}^{LD}(z\mid \widehat{f}(Az) = a)$ the corresponding conditional distribution on $\widehat{f}(Az) = a$ at $t_0$, with shorthand as $P_{t_0}^{LD}(a)$. Also give $P_z(z\mid \widehat{f}(Az) = a)$ a shorthand as $P^{LD}(a)$. In our theorems, $\mathcal{O}$ hides constant factors and higher order terms in $n_1^{-1}$ and $n_2^{-1}$ and , $\widetilde{\mathcal{O}}$ further hides logarithmic terms and can also hide factors in $d$.

### D.1   Parametric Conditional Score Matching Error

Theorems presented in Section 4 are established upon the conditional score estimation error, which has been studied in [8] for general distributions, but in Lemma D.1 we provide a new one specific to our setting where the true score is linear in input $(x_t,\widehat{y})$ due to the Gaussian design. Despite the linearity of score in Gaussian case, we emphasize matching score in (3.2) is not simply linear regression as $\mathcal{S}$ consists of an encoder-decoder structure for estimating matrix $A$ to reduce dimension (see §H for $\mathcal{S}$ construction and more proof details).

In the following lemma, we first present a general result for the case where the true score is within $\mathcal{S}$, which is constructed as a parametric function class. Then the score matching error is bounded in terms of $\mathcal{N}(\mathcal{S},1/n_1)$, the log covering number of $\mathcal{S}$, recall $n_1$ is the size of $\mathcal{D}_{\mathrm{unlabel}}$. Instantiating this general result, we derive score matching error for Gaussian case by upper bounding $\mathcal{N}(\mathcal{S},1/n_1)$ in this special case.

**Lemma D.1.**  Under Assumption 4.1, if $\nabla \log p_t(x\mid y)\in \mathcal{S}$, where

$$\mathcal{S} = \left\{ \mathbf{s}_{V,\psi}(x,y,t) = \frac{1}{h(t)}(V\cdot \psi(V^\top x,y,t) - x):\ V\in \mathbb{R}^{D\times d},\ \psi\in \Psi:\mathbb{R}^{d+1}\times[t_0,T]\to\mathbb{R}^d \right\},$$

$$((3.8)\ \text{revisited})$$

with $\Psi$ parametric. Then for $\delta\geq 0$, with probability $1-\delta$, the square score matching error is bounded by $\epsilon_{diff}^2 = \mathcal{O}\left(\frac{1}{t_0}\sqrt{\frac{\mathcal{N}(\mathcal{S},1/n_1)(d^2\vee D)\log\frac{1}{\delta}}{n_1}}\right)$, i.e.,

$$\frac{1}{T-t_0}\int_{t_0}^T \mathbb{E}_{(x_t,y)\sim P_t}\left[\|\nabla\log p_t(x_t|y) - \widehat{s}(x_t,y,t)\|_2^2\right]\mathrm{d}t \leq \epsilon_{diff}^2, \tag{D.1}$$

recall $P_t$ comes from $P_{x\widehat{y}}$ by noising $x$ at $t$ in the forward process. Under Assumption 4.4 and given $\widehat{f}(x) = \widehat{\theta}^\top x$ and $\widehat{y} = \widehat{f}(x) + \xi, \xi \sim \mathsf{N}(0, \nu^2)$, the score function $\nabla \log p_t(x \mid \widehat{y})$ to approximate is linear in $x$ and $\widehat{y}$. When approximated by $S$ with $\Psi$ linear, $\mathcal{N}(\mathcal{S}, 1/n_1) = \mathcal{O}((d^2 + Dd)\log(Ddn_1))$.

*Proof.* Proof is in §H. □

To provide fidelity and reward guarantees of $\widehat{P}_a$: the generated distribution of $x$ given condition $\widehat{y} = a$, we will need the following lemma. It provides a subspace recovery guarantee between $V$(score matching output) and $A$(ground truth), as well as a distance measure between distributions $P_a$ and $\widehat{P}_a$, given score matching error $\epsilon_{diff}$.

Note $P_a$ and $\widehat{P}_a$ are over $x$, which admits an underlying low-dimensional structure $x = Az$. Thus we measure distance between $P_a$ and $\widehat{P}_a$ by defining

**Definition D.2.** $TV(\widehat{P}_a) := \mathsf{d}_{\mathrm{TV}}\left(P_{t_0}^{LD}(z \mid \widehat{f}(Az) = a), (U^\top V^\top)_{\#}\widehat{P}_a\right)$ with notations:

- $\mathsf{d}_{\mathrm{TV}}(\cdot, \cdot)$ is the TV distance between two distribution.

- $f_\sharp P$ denotes a push-forward measure, i.e., for any measurable $\Omega$, $(f_\sharp P)(\Omega) = P(f^{-1}(\Omega))$

- $(V^\top)_{\#}\widehat{P}_a$ pushes generated $\widehat{P}_a$ forward to the low dimensional subspace using learned subspace matrix $V$. $U$ is an orthonormal matrix of dimension $d$.

- $P_{t_0}^{LD}(z \mid \widehat{f}(Az) = a)$ is close to $(A^\top)_{\#}P_a$, with $t_0$ taking account for the early stopping in backward process.

We note that there is a distribution shift between the training and the generated data, which has a profound impact on the generative performance. We quantify the influence of distribution shift by the following class restricted divergence measure.

**Definition D.3.** Distribution shift between two arbitrary distributions $P_1$ and $P_2$ restricted under function class $\mathcal{L}$ is defined as

$$\mathcal{T}(P_1, P_2; \mathcal{L}) = \sup_{l \in \mathcal{L}} \mathbb{E}_{x \sim P_1}[l(x)]/\mathbb{E}_{x \sim P_2}[l(x)] \quad \text{with arbitrary two distributions } P_1, P_2.$$

Definition D.3 is well perceived in bandit and RL literature [30, 29, 6, 12].

**Lemma D.4.** Given the square score matching error (D.1) upper bounded by $\epsilon_{diff}^2$, and when $P_z$ satisfying Assumption 4.3 with $c_0 I_d \preceq \mathbb{E}_{z \sim P_z}[zz^\top]$, it guarantees on for $x \sim \widehat{P}_a$ and $\angle(V, A) := \|VV^\top - AA^\top\|_{\mathrm{F}}^2$ that

$$(I_D - VV^\top)x \sim \mathsf{N}(0, \Lambda), \quad \Lambda \prec ct_0 I_D, \tag{D.2}$$

$$\angle(V, A) = \widetilde{\mathcal{O}}\left(\frac{t_0}{c_0} \cdot \epsilon_{diff}^2\right). \tag{D.3}$$

In addition,

$$TV(\widehat{P}_a) = \widetilde{\mathcal{O}}\left(\sqrt{\frac{\mathcal{T}(P(x, \widehat{y} = a), P_{x\widehat{y}}; \bar{\mathcal{S}})}{c_0}} \cdot \epsilon_{diff}\right). \tag{D.4}$$

with $\bar{\mathcal{S}} = \left\{\frac{1}{T - t_0}\int_{t_0}^T \mathbb{E}_{x_t|x}\|\nabla \log p_t(x_t \mid y) - s(x_t, y, t)\|_2^2 \mathrm{d}t : s \in \mathcal{S}\right\}$. $TV(\widehat{P}_a)$ and $\mathcal{T}(P(x, \widehat{y} = a), P_{x\widehat{y}}; \bar{\mathcal{S}})$ are defined in Definition D.2 and D.3.

*Proof.* Proof is in §E.2. □

## D.2 Proof of Theorem 4.5

*Proof.* **Proof of** $\angle(V, A)$**.** By Lemma 3 of [8], we have

$$\angle(V, A) = \mathcal{O}\left(\frac{t_0}{c_0} \cdot \epsilon_{diff}^2\right)$$

when the latent $z$ satisfying Assumption 4.3 and $c_0 I_d \preceq \mathbb{E}_{z \sim P_z}\left[zz^\top\right]$. Therefore, by (D.1), we have with high probability that

$$\angle(V, A) = \widetilde{\mathcal{O}}\left(\frac{1}{c_0}\sqrt{\frac{\mathcal{N}(\mathcal{S}, 1/n_1)(D \vee d^2)}{n_1}}\right).$$

When Assumption 4.4 holds, plugging in $c_0 = \lambda_{\min}$ and $\mathcal{N}(\mathcal{S}, 1/n_1) = \mathcal{O}((d^2 + Dd)\log(Ddn_1))$, it gives

$$\angle(V, A) = \widetilde{\mathcal{O}}\left(\frac{1}{\lambda_{\min}}\sqrt{\frac{(D \vee d^2)d^2 + (D \vee d^2)Dd}{n_1}}\right),$$

where $\widetilde{\mathcal{O}}$ hides logarithmic terms. When $D > d^2$, which is often the case in practical applications, we have

$$\angle(V, A) = \widetilde{\mathcal{O}}\left(\frac{1}{\lambda_{\min}}\sqrt{\frac{Dd^2 + D^2d}{n_1}}\right).$$

**Proof of** $\mathbb{E}_{x \sim \widehat{P}_a}[\|x_\perp\|_2]$**.** By the definition of $x_\perp$ that $x_\perp = (I_D - AA^\top)x$,

$$\mathbb{E}_{x \sim \widehat{P}_a}[\|x^\perp\|_2] = \mathbb{E}_{x \sim \widehat{P}_a}[\|(I_D - AA^\top)x\|_2] \leq \sqrt{\mathbb{E}_{x \sim \widehat{P}_a}[\|(I_D - AA^\top)x\|_2^2]}.$$

Score matching returns $V$ as an approximation of $A$, then

$$\|(I_D - AA^\top)x\|_2 \leq \|(I_D - VV^\top)x\|_2 + \|(VV^\top - AA^\top)x\|_2,$$

$$\mathbb{E}_{x \sim \widehat{P}_a}[\|(I_D - AA^\top)x\|_2^2] \leq 2\mathbb{E}_{x \sim \widehat{P}_a}[\|(I_D - VV^\top)x\|_2^2] + 2\mathbb{E}_{x \sim \widehat{P}_a}[\|(VV^\top - AA^\top)x\|_2^2],$$

where by (D.2) in Lemma D.4 we have

$$(I_D - VV^\top)x \sim \mathsf{N}(0, \Lambda), \quad \Lambda \prec \mathsf{ct_0 I}$$

for some constant $c \geq 0$. Thus

$$\mathbb{E}_{x \sim \widehat{P}_a}\left[\|(I_D - VV^\top)x\|_2^2\right] = \mathrm{Tr}(\Lambda) \leq ct_0 D. \tag{D.5}$$

On the other hand,

$$\|(VV^\top - AA^\top)x\|_2^2 \leq \|VV^\top - AA^\top\|_{op}^2\|x\|_2^2 \leq \|VV^\top - AA^\top\|_F^2\|x\|_2^2,$$

where $\|VV^\top - AA^\top\|_F^2$ has an upper bound as in (D.3) and $\mathbb{E}_{x \sim \widehat{P}_a}\left[\|x\|_2^2\right])$ is bounded in Lemma E.3 by

$$\mathbb{E}_{x \sim \widehat{P}_a}\left[\|x\|_2^2\right] = \mathcal{O}\left(ct_0 D + M(a) \cdot (1 + TV(\widehat{P}_a))\right).$$

with $M(a) = O\left(\frac{a^2}{\|\beta^*\|_\Sigma} + d\right)$.

Therefore, to combine things together, we have

$$\mathbb{E}_{x \sim \widehat{P}_a}[\|x^\perp\|_2] \leq \sqrt{2\mathbb{E}_{x \sim \widehat{P}_a}[\|(I_D - VV^\top)x\|_2^2] + 2\mathbb{E}_{x \sim \widehat{P}_a}[\|(VV^\top - AA^\top)x\|_2^2]}$$

$$\leq c'\sqrt{t_0 D} + 2\sqrt{\angle(V, A)} \cdot \sqrt{\mathbb{E}_{x \sim \widehat{P}_a}\left[\|x\|_2^2\right]}$$

$$= \mathcal{O}\left(\sqrt{t_0 D} + \sqrt{\angle(V, A)} \cdot \sqrt{M(a)}\right).$$

$\mathcal{O}$ hides multiplicative constant and $\sqrt{\angle(V, A)t_0 D}$, $\sqrt{\angle(V, A)M(a)TV(\widehat{P}_a)}$, which are terms with higher power of $n_1^{-1}$ than the leading term. $\qquad\square$

**Remark of Theorem 4.5.** 1. Guarantee (4.2) applies to general distributions with light tail as assumed in Assumption 4.3.

2. Guarantee (4.3) guarantees high fidelity of generated data in terms of staying in the subspace when we have access to a large unlabeled dataset.

3. Guarantee (4.3) shows that $\mathbb{E}_{x \sim \widehat{P}_a}[\|x_\perp\|_2]$ scales up when $t_0$ goes up, which aligns with the dynamic in backward process that samples are concentrating to the learned subspace as $t_0$ goes to 0. Taking $t_0 \to 0$, $\mathbb{E}_{x \sim \widehat{P}_a}[\|x_\perp\|]$ has the decay in $O(n_1^{-\frac{1}{4}})$. However, taking $t_0 \to 0$ is not ideal for the sake of high reward of $x$, we take the best trade-off of $t_0$ later in Theorem 4.6.

## D.3 Proof of Theorem 4.6

Proof of Theorem 4.6 and that of some results in "Implications and Discussions" following the theorem in main paper are provided in this section. This section breaks down into three parts: **Suboptimality Decomposition**, **Bounding $\mathcal{E}_1$ Relating to Offline Bandits**, **Bounding $\mathcal{E}_2$ and the Distribution Shift in Diffusion**.

### D.3.1 $\texttt{SubOpt}(\widehat{P}_a; y^* = a)$ Decomposition

*Proof.* Recall notations $\widehat{P}_a := \widehat{P}(\cdot|\widehat{y} = a)$ (generated distribution) and $P_a := P(\cdot|\widehat{y} = a)$ (target distribution) and $f^*(x) = g^*(x_\parallel) - h^*(x_\perp)$. $\mathbb{E}_{x \sim \widehat{P}_a}[f^*(x)]$ can be decomposed into 3 terms:

$$
\begin{aligned}
\mathbb{E}_{x \sim \widehat{P}_a}[f^\star(x)] \geq & \mathbb{E}_{x \sim P_a}[f^\star(x)] - \left| \mathbb{E}_{x \sim \widehat{P}_a}[f^*(x)] - \mathbb{E}_{x \sim P_a}[f^\star(x)] \right| \\
\geq & \mathbb{E}_{x \sim P_a}[\widehat{f}(x)] - \mathbb{E}_{x \sim P_a}\left[\left|\widehat{f}(x) - f^*(x)\right|\right] - \left| \mathbb{E}_{x \sim \widehat{P}_a}[f^*(x)] - \mathbb{E}_{x \sim P_a}[f^\star(x)] \right| \\
\geq & \mathbb{E}_{x \sim P_a}[\widehat{f}(x)] - \underbrace{\mathbb{E}_{x \sim P_a}\left[\left|\widehat{f}(x) - g^*(x)\right|\right]}_{\mathcal{E}_1} \\
& - \underbrace{\left| \mathbb{E}_{x \sim P_a}[g^*(x_\parallel)] - \mathbb{E}_{x \sim \widehat{P}_a}[g^*(x_\parallel)] \right|}_{\mathcal{E}_2} - \underbrace{\mathbb{E}_{x \sim \widehat{P}_a}[h^*(x_\perp)]}_{\mathcal{E}_3},
\end{aligned}
$$

where $\mathbb{E}_{x \sim P_a}[\widehat{f}(x)] = \mathbb{E}_{a \sim q}[a]$ and we use $x = x_\parallel$, $f^*(x) = g^*(x)$ when $x \sim P_a$. Therefore

$$
\begin{aligned}
\texttt{SubOpt}(\widehat{P}_a; y^* = a) = & \; a - \mathbb{E}_{x \sim \widehat{P}_a}[f^\star(x)] \\
\leq & \underbrace{\mathbb{E}_{x \sim P_a}\left[\left|(\widehat{\theta} - \theta^*)^\top x\right|\right]}_{\mathcal{E}_1} + \underbrace{\left| \mathbb{E}_{x \sim P_a}[g^*(x_\parallel)] - \mathbb{E}_{x \sim \widehat{P}_a}[g^*(x_\parallel)] \right|}_{\mathcal{E}_2} \\
& + \underbrace{\mathbb{E}_{x \sim \widehat{P}_a}[h^*(x_\perp)]}_{\mathcal{E}_3}.
\end{aligned}
$$

$\square$

$\mathcal{E}_1$ comes from regression: prediction/generalization error onto $P_a$, which is independent from any error of distribution estimation that occurs in diffusion. $\mathcal{E}_2$ and $\mathcal{E}_3$ do not measure regression-predicted $\widehat{f}$, thus they are independent from the prediction error in $\widehat{f}$ for pseudo-labeling. $\mathcal{E}_2$ measures the disparity between $\widehat{P}_a$ and $P_a$ on the subspace support and $\mathcal{E}_3$ measures the off-subspace component in generated $\widehat{P}_a$.

### D.3.2 Bounding $\mathcal{E}_1$ Relating to Offline Bandits

For all $x_i \in \mathcal{D}_{\text{label}}$, $y_i = f^*(x_i) + \epsilon_i = g(x_i) + \epsilon_i$. Thus, trained on $\mathcal{D}_{\text{label}}$ the prediction model $\widehat{f}$ is essentially approximating $g$. By estimating $\theta^*$ with ridge regression on $\mathcal{D}_{\text{label}}$, we have $\widehat{f}(x) = \widehat{\theta}^\top x$ with

$$
\widehat{\theta} = \left(X^\top X + \lambda I\right)^{-1} X^\top \left(X\theta^* + \eta\right), \tag{D.6}
$$

where $X^\top = (x_1, \cdots, x_i, \cdots, x_{n_2})$ and $\eta = (\epsilon_1, \cdots, \epsilon_i, \cdots, \epsilon_{n_2})$.

**Lemma D.5.** Under Assumption 4.1 and 4.2 and given $\epsilon_i \sim \mathsf{N}(0, \sigma^2)$, define $V_\lambda := X^\top X + \lambda I$, $\widehat{\Sigma}_\lambda := \frac{1}{n_2} V_\lambda$ and $\Sigma_{P_a} := \mathbb{E}_{x \sim P_a} x x^\top$ the covariance matrix (uncentered) of $P_a$, and take $\lambda = 1$, then with high probability

$$\mathcal{E}_1 \leq \sqrt{\mathrm{Tr}(\widehat{\Sigma}_\lambda^{-1} \Sigma_{P_a})} \cdot \frac{\mathcal{O}\left(\sqrt{d \log n_2}\right)}{\sqrt{n_2}}. \tag{D.7}$$

*Proof.* Proof is in §E.3. $\qquad\square$

**Lemma D.6.** Under Assumption 4.1, 4.2 and 4.4, when $\lambda = 1$, $P_a$ has a shift from the empirical marginal of $x$ in dataset by

$$\mathrm{Tr}(\widehat{\Sigma}_\lambda^{-1} \Sigma_{P_a}) \leq \mathcal{O}\left(\frac{a^2}{\|\beta^*\|_\Sigma} + d\right). \tag{D.8}$$

when $n_2 = \Omega(\max\{\frac{1}{\lambda_{\min}}, \frac{d}{\|\beta^*\|_\Sigma^2}\})$.

*Proof.* Proof is in §E.4. $\qquad\square$

### D.3.3 Bounding $\mathcal{E}_2$ and the Distribution Shift in Diffusion

**Lemma D.7.** Under Assumption 4.1, 4.2 and 4.4, when $t_0 = \left((Dd^2 + D^2d)/n_1\right)^{1/6}$

$$\mathcal{E}_2 = \widetilde{\mathcal{O}}\left(\sqrt{\frac{\mathcal{T}(P(x, \widehat{y} = a), P_{x\widehat{y}}; \bar{\mathcal{S}})}{\lambda_{\min}}} \cdot \left(\frac{Dd^2 + D^2d}{n_1}\right)^{\frac{1}{6}} \cdot a\right).$$

*Proof.* Proof is in §E.5. $\qquad\square$

Note that $\mathcal{T}(P(x, \widehat{y} = a), P_{x\widehat{y}}; \bar{\mathcal{S}})$ depends on $a$ and measures the distribution shift between the desired distribution $P(x, \widehat{y} = a)$ and the data distribution $P_{x\widehat{y}}$. To understand this distribution's dependency on $a$, it what follows we give $\mathcal{T}(P(x, \widehat{y} = a), P_{x\widehat{y}}; \bar{\mathcal{S}})$ a shorthand as $\mathtt{DistroShift}^2(a)$ and give it an upper bound in one special case of the problem.

**Distribution Shift** In the special case of covariance $\Sigma$ of $z$ is known and $\|A - V\|_2^2 = \mathcal{O}\left(\|AA^\top - VV^\top\|_F^2\right)$, we showcase a bound on the distribution shift in $\mathcal{E}_2$, as promised in the discussion following Theorem 4.6. We have

$$\mathtt{DistroShift}^2(a) = \frac{\mathbb{E}_{P_{x,\widehat{y}=a}}[\ell(x, y; \widehat{s})]}{\mathbb{E}_{P_{x\widehat{y}}}[\ell(x, y; \widehat{s})]},$$

where $\ell(x, y; \widehat{s}) = \frac{1}{T - t_0} \int_{t_0}^T \mathbb{E}_{x'|x} \|\nabla_{x'} \log \phi_t(x'|x) - \widehat{s}(x', y, t)\|_2^2 dt$. By Proposition 3.1, it suffices to bound

$$\mathtt{DistroShift}^2(a) = \frac{\mathbb{E}_{P_{x,\widehat{y}=a}}[\int_{t_0}^T \|\nabla \log p_t(x, y) - \widehat{s}(x, y, t)\|_2^2 dt]}{\mathbb{E}_{P_{x\widehat{y}}}[\int_{t_0}^T \|\nabla \log p_t(x, y) - \widehat{s}(x, y, t)\|_2^2 dt]}.$$

We expand the difference $\|\nabla \log p_t(x, y) - \widehat{s}(x, y, t)\|_2^2$ by

$$\|\nabla \log p_t(x, y) - \widehat{s}(x, y, t)\|_2^2 \leq \frac{2}{h^2(t)} \left[\|(A - V)B_t(A^\top x + \nu^{-2} y \theta)\|_2^2 + \|V B_t(A - V)^\top x\|_2^2\right]$$

$$\leq \frac{2}{h^2(t)} \left[\|A - V\|_2^2 \|B_t(A^\top x + \nu^{-2} y \theta)\|_2^2 + \|A - V\|_2^2 \|x\|_2^2\right]$$

$$\leq \frac{2}{h^2(t)} \|A - V\|_2^2 (3\|x\|_2^2 + y^2),$$

where we recall $B_t$ is defined in (H.2) and in the last inequality, we use $(a + b)^2 \leq 2a^2 + 2b^2$. In the case of covariance matrix $\Sigma$ is known, i.e., $B_t$ is known, we also consider matrix $V$ directly matches $A$ without rotation. Then by [8, Lemma 3 and 17], we have $\|A - V\|_2^2 = \mathcal{O}\left(\|AA^\top - VV^\top\|_F^2\right) =$

$\mathcal{O}\left(t_0/c_0 \mathbb{E}_{P_{x\widehat{y}}}[\ell(x,y;\widehat{s})]\right)$. To this end, we only need to find $\mathbb{E}_{P_{x|\widehat{y}=a}}[\|x\|_2^2]$. Since we consider on-support $x$, which can be represented as $x = Az$, we have $\|x\|_2 = \|z\|_2$. Thus, we only need to find the conditional distribution of $z|\widehat{y} = a$. Fortunately, we know $(z, \widehat{y})$ is jointly Gaussian, with mean 0 and covariance

$$\begin{bmatrix} \Sigma & \Sigma\widehat{\beta} \\ \widehat{\beta}^\top\Sigma & \widehat{\beta}^\top\Sigma\widehat{\beta} + \nu^2 \end{bmatrix}.$$

Consequently, the conditional distribution of $z|\widehat{y} = a$ is still Gaussian, with mean $\Sigma\widehat{\beta}a/(\widehat{\beta}^\top\Sigma\widehat{\beta} + \nu^2)$ and covariance $\Sigma - \Sigma\widehat{\beta}\widehat{\beta}^\top\Sigma/(\widehat{\beta}^\top\Sigma\widehat{\beta} + \nu^2)$. Hence, we have

$$\mathbb{E}_{P_{z|\widehat{y}=a}}[\|z\|_2^2] = \frac{1}{(\widehat{\beta}^\top\Sigma\widehat{\beta} + \nu^2)^2}\left((a^2 - \widehat{\beta}^\top\Sigma\widehat{\beta} - \nu^2)\widehat{\beta}^\top\Sigma^2\widehat{\beta}\right) + \mathrm{Tr}(\Sigma) = \mathcal{O}\left(a^2 \vee d\right).$$

We integrate over $t$ for the numerator in $\texttt{DistroShift}(a)$ to obtain $\mathbb{E}_{P_{x|\widehat{y}=a}}[\int_{t_0}^T \|\nabla\log p_t(x,y) - \widehat{s}(x,y,t)\|_2^2 dt = \mathcal{O}\left((a^2 \vee d)\frac{1}{c_0}\mathbb{E}_{P_{x\widehat{y}}}[\ell(x,y;\widehat{s})]\right)$. Note the cancellation between the numerator and denominator, we conclude

$$\texttt{DistroShift}(a) = \mathcal{O}\left(\frac{1}{c_0}(a \vee \sqrt{d})\right).$$

As $d$ is a natural upper bound of $\sqrt{d}$ and viewing $c_0$ as a constant, we have $\texttt{DistroShift}(a) = \mathcal{O}\left(a \vee d\right)$ as desired.

# E  Supporting lemmas and proofs

## E.1  Supporting lemmas

**Lemma E.1.** The estimated subspace $V$ satisfies

$$\|VU - A\|_F = \mathcal{O}\left(d^{\frac{3}{2}}\sqrt{\angle(V,A)}\right) \tag{E.1}$$

for some orthogonal matrix $U \in \mathbb{R}^{d\times d}$.

*Proof.* Proof is in §E.6. $\qquad\square$

**Lemma E.2.** Suppose $P_1$ and $P_2$ are two distributions over $\mathbb{R}^d$ and $m$ is a function defined on $\mathbb{R}^d$, then $|\mathbb{E}_{x\sim P_1}[m(z)] - \mathbb{E}_{z\sim P_2}[m(z)]|$ can be bounded in terms of $\mathrm{d}_{\mathrm{TV}}(P_1, P_2)$, specifically when $P_1$ and $P_2$ are Gaussians and $m(z) = \|z\|_2^2$:

$$\mathbb{E}_{x\sim P_1}[\|z\|_2^2] = \mathcal{O}\left(\mathbb{E}_{z\sim P_2}[\|z\|_2^2](1 + \mathrm{d}_{\mathrm{TV}}(P_1, P_2))\right). \tag{E.2}$$

When $P_1$ and $P_2$ are Gaussians and $m(z) = \|z\|_2$:

$$|\mathbb{E}_{z\sim P_1}[\|z\|_2] - \mathbb{E}_{z\sim P_2}[\|z\|_2]| = \mathcal{O}\left(\left(\sqrt{\mathbb{E}_{z\sim P_1}[\|z\|_2^2]} + \sqrt{\mathbb{E}_{z\sim P_2}[\|z\|_2^2]}\right)\cdot\mathrm{d}_{\mathrm{TV}}(P_1, P_2)\right). \tag{E.3}$$

*Proof.* Proof is in §E.7. $\qquad\square$

**Lemma E.3.** We compute $\mathbb{E}_{z\sim P^{LD}(a)}[\|z\|_2^2], \mathbb{E}_{z\sim P_{t_0}^{LD}(a)}[\|z\|_2^2], \mathbb{E}_{x\sim\widehat{P}_a}[\|x\|_2^2], \mathbb{E}_{z\sim(U^\top V^\top)_\#\widehat{P}_a}[\|z\|_2^2]$ in this Lemma.

$$\mathbb{E}_{z\sim P^{LD}(a)}[\|z\|_2^2] = \frac{\widehat{\beta}^\top\Sigma^2\widehat{\beta}}{\left(\|\widehat{\beta}\|_\Sigma^2 + \nu^2\right)^2}a^2 + \mathrm{trace}(\Sigma - \Sigma\widehat{\beta}\left(\widehat{\beta}^\top\Sigma\widehat{\beta} + \nu^2\right)^{-1}\widehat{\beta}^\top\Sigma). \tag{E.4}$$

Let $M(a) := \mathbb{E}_{z\sim P^{LD}(a)}[\|z\|_2^2]$, which has an upper bound $M(a) = O\left(\frac{a^2}{\|\beta^*\|_\Sigma} + d\right)$.

$$\mathbb{E}_{z\sim P_{t_0}^{LD}(a)}[\|z\|_2^2] \leq M(a) + t_0 d. \tag{E.5}$$

$$\mathbb{E}_{x\sim\widehat{P}_a}\left[\|x\|_2^2\right] \leq \mathcal{O}\left(ct_0 D + M(a)\cdot(1 + TV(\widehat{P}_a))\right). \tag{E.6}$$

*Proof.* Proof is in §E.8. $\qquad\square$

## E.2 Proof of Lemma D.4

*Proof.* The first two assertions (D.2) and (D.3) are consequences of [8, Theorem 3, item 1 and 3]. To show (D.4), we first have the conditional score matching error under distribution shift being

$$\mathcal{T}(P(x, \widehat{y} = a), P_{x\widehat{y}}; \bar{\mathcal{S}}) \cdot \epsilon_{diff}^2,$$

where $\mathcal{T}(P(x, \widehat{y} = a), P_{x\widehat{y}}; \bar{\mathcal{S}})$ accounts for the distribution shift as in the parametric case (Lemma D.7). Then we apply [8, Theorem 3, item 2] to conclude

$$TV(\widehat{P}_a) = \widetilde{\mathcal{O}}\left(\sqrt{\frac{\mathcal{T}(P(x, \widehat{y} = a), P_{x\widehat{y}}; \bar{\mathcal{S}})}{c_0}} \cdot \epsilon_{diff}\right).$$

The proof is complete. $\square$

## E.3 Proof of Lemma D.5

*Proof.* Given

$$\mathcal{E}_1 = \mathbb{E}_{\widehat{P}_a}\left|x^\top(\theta^* - \widehat{\theta})\right| \leq \mathbb{E}_{\widehat{P}_a}\|x\|_{V_\lambda^{-1}} \cdot \|\theta^* - \widehat{\theta}\|_{V_\lambda},$$

then things to prove are

$$\mathbb{E}_{\widehat{P}_a}\|x\|_{V_\lambda^{-1}} = \sqrt{\text{trace}(V_\lambda^{-1}\Sigma_{\widehat{P}_a})}; \tag{E.7}$$

$$\|\theta^* - \widehat{\theta}\|_{V_\lambda} \leq \mathcal{O}\left(\sqrt{d\log n_2}\right), \tag{E.8}$$

where the second inequality is to be proven with high probability w.r.t the randomness in $\mathcal{D}_{label}$. For (E.7), $\mathbb{E}_{\widehat{P}_a}\|x\|_{V_\lambda^{-1}} \leq \sqrt{\mathbb{E}_{\widehat{P}_a} x^\top V_\lambda^{-1} x} = \sqrt{\mathbb{E}_{\widehat{P}_a} \text{trace}(V_\lambda^{-1} x x^\top)} = \sqrt{\text{trace}(V_\lambda^{-1} \mathbb{E}_{\widehat{P}_a} x x^\top)}$.

For (E.8), what's new to prove compared to a classic bandit derivation is its $d$ dependency instead of $D$, due to the linear subspace structure in $x$. From the closed form solution of $\widehat{\theta}$, we have

$$\widehat{\theta} - \theta^* = V_\lambda^{-1} X^\top \eta - \lambda V_\lambda^{-1} \theta^*. \tag{E.9}$$

Therefore,

$$\|\theta^* - \widehat{\theta}\|_{V_\lambda} \leq \|X^\top \eta\|_{V_\lambda^{-1}} + \lambda\|\theta^*\|_{V_\lambda^{-1}}, \tag{E.10}$$

where $\lambda\|\theta^*\|_{V_\lambda^{-1}} \leq \sqrt{\lambda}\|\theta^*\|_2 \leq \sqrt{\lambda}$ and

$$\|X^\top \eta\|_{V_\lambda^{-1}}^2 = \eta^\top X \left(X^\top X + \lambda I_D\right)^{-1} X^\top \eta$$

$$= \eta^\top X X^\top \left(X X^\top + \lambda I_{n_2}\right)^{-1} \eta.$$

Let $Z^\top = (z_1, \cdots, z_i, \cdots, z_{n_2})$ s.t. $A z_i = x_i$, then it holds that $X = Z A^\top$, and $X X^\top = Z A^\top A Z^\top = Z Z^\top$, thus

$$\|X^\top \eta\|_{V_\lambda^{-1}}^2 = \eta^\top X X^\top \left(X X^\top + \lambda I_{n_2}\right)^{-1} \eta$$

$$= \eta^\top Z Z^\top \left(Z Z^\top + \lambda I_{n_2}\right)^{-1} \eta$$

$$= \eta^\top Z \left(Z^\top Z + \lambda I_d\right)^{-1} Z^\top \eta$$

$$= \|Z^\top \eta\|_{(Z^\top Z + \lambda I_d)^{-1}}.$$

With probability $1 - \delta$, $\|z_i\|^2 \leq d + \sqrt{d\log\left(\frac{2n_2}{\delta}\right)} := L^2, \forall i \in [n_2]$. Then Theorem 1 in "Improved algorithms for linear stochastic bandits" (by Yasin Abbasi-Yadkori, David Pal, and Csaba Szepesvari) gives rise to

$$\|Z^\top \eta\|_{(Z^\top Z + \lambda I_d)^{-1}} \leq \sqrt{2\log(2/\delta) + d\log(1 + n_2 L^2/(\lambda d))}$$

with probability $1 - \delta/2$. Combine things together and plugging in $\lambda = 1$, $L^2 = d + \sqrt{d\log\left(\frac{2n_2}{\delta}\right)}$, we have with high probability

$$\|\theta^* - \widehat{\theta}\|_{V_\lambda} = \mathcal{O}\left(\sqrt{d\log\left(n_2\sqrt{\log(n_2)}\right)}\right) = \mathcal{O}\left(\sqrt{d\log n_2 + \frac{1}{2}d\log(\log n_2)}\right) = \mathcal{O}\left(\sqrt{d\log n_2}\right).$$

$\square$

### E.4 Proof of Lemma D.6

*Proof.* Recall the definition of $\widehat{\Sigma}_\lambda$ and $\Sigma_{P_a}$ that

$$\widehat{\Sigma}_\lambda = \frac{1}{n_2} X^\top X + \frac{\lambda}{n_2} I_D,$$

$$\Sigma_{P_a} = \mathbb{E}_{x \sim P_a} \left[ xx^\top \right],$$

where $X$ are stack matrix of data supported on $\mathcal{A}$ and $P_a$ is also supported on $\mathcal{A}$, $\mathcal{A}$ is the subspace encoded by matrix $A$. The following lemma shows it is equivalent to measure $\text{trace}(\widehat{\Sigma}_\lambda^{-1}\Sigma_{P_a})$ on $\mathcal{A}$ subspace.

**Lemma E.4.** For any P.S.D. matrices $\Sigma_1, \Sigma_2 \in \mathbb{R}^{d \times d}$ and $A \in \mathbb{R}^{D \times d}$ such that $A^\top A = I_d$, we have

$$\text{Tr}\left( (\lambda I_D + A\Sigma_1 A^\top)^{-1} A\Sigma_2 A^\top \right) = \text{Tr}\left( (\lambda I_d + \Sigma_1)^{-1} \Sigma_2 \right).$$

The lemma above allows us to abuse notations $\widehat{\Sigma}_\lambda$ and $\Sigma_{P_a}$ in the following way while keeping the same $\text{trace}(\widehat{\Sigma}_\lambda^{-1}\Sigma_{P_a})$ value:

$$\widehat{\Sigma}_\lambda = \frac{1}{n_2} Z^\top Z + \frac{\lambda}{n_2} I_d,$$

$$\Sigma_{P_a} = \mathbb{E}_{z \sim P^{LD}(a)} \left[ zz^\top \right],$$

where $Z^\top = (z_1, \cdots, z_i, \cdots, z_{n_2})$ s.t. $Az_i = x_i$ and recall notataion $P^{LD}(a) = P_z\left( z \mid \widehat{f}(Az) \right)$.

Given $z \sim \mathsf{N}(\mu, \Sigma)$, as a proof artifact, let $\widehat{f}(x) = \widehat{\theta}^\top x + \xi, \xi \sim \mathsf{N}(0, \nu^2)$ where we will let $\nu \to 0$ in the end, then let $\widehat{\beta} = A^\top \widehat{\theta} \in \mathbb{R}^d$, $(z, \widehat{f}(Az))$ has a joint distribution

$$(z, \widehat{f}) \sim \mathsf{N}\left( \begin{bmatrix} \mu \\ \widehat{\beta}^\top \mu \end{bmatrix}, \begin{bmatrix} \Sigma & \Sigma\widehat{\beta} \\ \widehat{\beta}^\top \Sigma & \widehat{\beta}^\top \Sigma \widehat{\beta} + \nu^2 \end{bmatrix} \right). \tag{E.11}$$

Then we have the conditional distribution $z \mid \widehat{f}(Az) = a$ following

$$P_z\left( z \mid \widehat{f}(Az) = a \right) = \mathsf{N}\left( \mu + \Sigma\widehat{\beta}\left( \widehat{\beta}^\top \Sigma\widehat{\beta} + \nu^2 \right)^{-1}(a - \widehat{\beta}^\top \mu), \Gamma \right) \tag{E.12}$$

with $\Gamma := \Sigma - \Sigma\widehat{\beta}\left( \widehat{\beta}^\top \Sigma\widehat{\beta} + \nu^2 \right)^{-1}\widehat{\beta}^\top \Sigma$.

When $\mu = 0$, we compute $\text{trace}(\widehat{\Sigma}_\lambda^{-1}\Sigma_{P_a})$ as

$$\text{trace}(\widehat{\Sigma}_\lambda^{-1}\Sigma_{P_a}) = \text{trace}\left( \widehat{\Sigma}_\lambda^{-1} \frac{\Sigma\widehat{\beta}\widehat{\beta}^\top \Sigma}{\left( \|\widehat{\beta}\|_\Sigma^2 + \nu^2 \right)^2} a^2 \right) + \text{trace}\left( \widehat{\Sigma}_\lambda^{-1} \Gamma \right)$$

$$= \text{trace}\left( \frac{\widehat{\beta}^\top \Sigma\widehat{\Sigma}_\lambda^{-1}\Sigma\widehat{\beta}}{\left( \|\widehat{\beta}\|_\Sigma^2 + \nu^2 \right)^2} a^2 \right) + \text{trace}\left( \widehat{\Sigma}_\lambda^{-1}\Sigma \right) - \text{trace}\left( \widehat{\Sigma}_\lambda^{-1} \frac{\Sigma\widehat{\beta}\widehat{\beta}^\top \Sigma}{\|\widehat{\beta}\|_\Sigma^2 + \nu^2} \right)$$

$$= \text{trace}\left( \frac{\Sigma^{1/2}\widehat{\beta}\widehat{\beta}^\top \Sigma^{1/2}\Sigma^{1/2}\widehat{\Sigma}_\lambda^{-1}\Sigma^{1/2}}{\left( \|\widehat{\beta}\|_\Sigma^2 + \nu^2 \right)^2} a^2 \right)$$

$$\leq \frac{\|\Sigma\widehat{\Sigma}_\lambda^{-1}\Sigma\|_{op} \cdot \|\widehat{\beta}\|_\Sigma^2}{\left( \|\widehat{\beta}\|_\Sigma^2 + \nu^2 \right)^2} \cdot a^2 + \text{trace}\left( \Sigma^{\frac{1}{2}}\widehat{\Sigma}_\lambda^{-1}\Sigma^{\frac{1}{2}} \right)$$

By the Lemma 3 in [7], it holds that

$$\| \Sigma^{\frac{1}{2}}\widehat{\Sigma}_\lambda^{-1}\Sigma^{\frac{1}{2}} - I_d \|_2 \leq O\left( \frac{1}{\sqrt{\lambda_{\min} n_2}} \right). \tag{E.13}$$

Therefore,

$$\text{trace}(\widehat{\Sigma}_\lambda^{-1} \Sigma_{P_a}) \le \frac{1 + \frac{1}{\sqrt{\lambda_{\min} n_2}}}{\|\widehat{\beta}\|_\Sigma^2} \cdot a^2 + O\left(d\left(1 + \frac{1}{\sqrt{\lambda_{\min} n_2}}\right)\right).$$

Then, what left is to bound $\|\widehat{\beta}\|_\Sigma = \|\widehat{\theta}\|_{A\Sigma A^\top} \ge \|\theta^*\|_{A\Sigma A^\top} - \|\widehat{\theta} - \theta^*\|_{A\Sigma A^\top}$ by triangle inequality. On one hand,

$$\|\theta^*\|_{A\Sigma A^\top} = \|\beta^*\|_\Sigma. \tag{E.14}$$

On the other hand,

$$\|\widehat{\theta} - \theta^*\|_{A\Sigma A^\top} = \mathcal{O}\left(\|\widehat{\theta} - \theta^*\|_{\widehat{\Sigma}_\lambda}\right)$$

$$= \mathcal{O}\left(\frac{\|\widehat{\theta} - \theta^*\|_{V_\lambda}}{\sqrt{n_2}}\right)$$

$$= \mathcal{O}\left(\sqrt{\frac{d\log(n_2)}{n_2}}\right).$$

with high probability. Thus when $n_2 = \Omega(\frac{d}{\|\beta^*\|_\Sigma^2})$

$$\|\widehat{\beta}\|_\Sigma \ge \frac{1}{2}\|\beta^*\|_\Sigma.$$

Therefore

$$\text{trace}(\widehat{\Sigma}_\lambda^{-1} \Sigma_{P_a}) \le \mathcal{O}\left(\frac{1 + \frac{1}{\sqrt{\lambda_{\min} n_2}}}{\|\beta^*\|_\Sigma} \cdot a^2 + d\left(1 + \frac{1}{\sqrt{\lambda_{\min} n_2}}\right)\right) = \mathcal{O}\left(\frac{a^2}{\|\beta^*\|_\Sigma} + d\right). \tag{E.15}$$

when $n_2 = \Omega(\max\{\frac{1}{\lambda_{\min}}, \frac{d}{\|\beta^*\|_\Sigma^2}\})$. $\qquad\square$

### E.5 Proof of lemma D.7

*Proof.* Recall the definition of $g(x)$ that

$$g(x) = \theta^{*\top} A A^\top x,$$

note that $g(x) = \theta^{*\top} x$ when $x$ is supported on $\mathcal{A}$. Thus,

$$\left|\mathbb{E}_{x \sim P_a}[g(x)] - \mathbb{E}_{x \sim \widehat{P}_a}[g(x)]\right|$$

$$= \left|\mathbb{E}_{x \sim P_a}[\theta^{*\top} x] - \mathbb{E}_{x \sim \widehat{P}_a}[\theta^{*\top} A A^\top x]\right|$$

$$\le \left|\mathbb{E}_{x \sim P_a}[\theta^{*\top} x] - \mathbb{E}_{x \sim \widehat{P}_a}[\theta^{*\top} V V^\top x]\right| + \underbrace{\left|\mathbb{E}_{x \sim \widehat{P}_a}[\theta^{*\top} V V^\top x] - \mathbb{E}_{x \sim \widehat{P}_a}[\theta^{*\top} A A^\top x]\right|}_{e_1},$$

where

$$e_1 = \left|\mathbb{E}_{x \sim \widehat{P}_a}[\theta^{*\top}\left(V V^\top - A A^\top\right) x]\right|$$

$$\le \mathbb{E}_{x \sim \widehat{P}_a}\left[\left(\left\|\left(V V^\top - A A^\top\right) x\right\|\right)\right]$$

$$\le \|V V^\top - A A^\top\|_F \cdot \sqrt{\mathbb{E}_{x \sim \widehat{P}_a}\left[\|x\|_2^2\right]}.$$

Use notation $P^{LD}(a) = P(z \mid \widehat{f}(Az) = a)$, $P_{t_0}^{LD}(a) = P_{t_0}(z \mid \widehat{f}(Az) = a)$

$$\left|\mathbb{E}_{x \sim P_a}[\theta^{*\top} x] - \mathbb{E}_{x \sim \widehat{P}_a}[\theta^{*\top} V V^\top x]\right|$$

$$= \left|\mathbb{E}_{z \sim P(z|\widehat{f}(Az)=a)}[\theta^{*\top} Az] - \mathbb{E}_{z \sim (U^\top V^\top)_\# \widehat{P}_a}[\theta^{*\top} V U z]\right|$$

$$\le \left|\mathbb{E}_{z \sim P_{t_0}^{LD}(a)}[\theta^{*\top} Az] - \mathbb{E}_{z \sim (U^\top V^\top)_\# \widehat{P}_a}[\theta^{*\top} V U z]\right| + \underbrace{\left|\mathbb{E}_{z \sim P_{t_0}^{LD}(a)}[\theta^{*\top} Az] - \mathbb{E}_{z \sim P^{LD}(a)}[\theta^{*\top} Az]\right|}_{e_2},$$

here
$$e_2 = \left| \alpha(t_0)\mathbb{E}_{z\sim P^{LD}(a)}[\theta^{*\top}Az] + h(t_0)\mathbb{E}_{u\sim \mathsf{N}(0,\mathsf{I_d})}[\theta^{*\top}Au] - \mathbb{E}_{z\sim P^{LD}(a)}[\theta^{*\top}Az] \right|$$
$$\leq (1-\alpha(t_0))\left| \mathbb{E}_{z\sim P^{LD}(a)}[\theta^{*\top}Az] \right|$$
$$\leq t_0 \cdot \mathbb{E}_{z\sim P^{LD}(a)}[\|z\|_2].$$

Then what is left to bound is
$$\left| \mathbb{E}_{z\sim P^{LD}_{t_0}(a)}[\theta^{*\top}Az] - \mathbb{E}_{z\sim (U^\top V^\top)_\#\widehat{P}_a}[\theta^{*\top}VUz] \right|$$
$$\leq \underbrace{\left| \mathbb{E}_{z\sim P^{LD}_{t_0}(a)}[\theta^{*\top}VUz] - \mathbb{E}_{z\sim (U^\top V^\top)_\#\widehat{P}_a}[\theta^{*\top}VUz] \right|}_{e_3} + \underbrace{\left| \mathbb{E}_{z\sim P^{LD}_{t_0}(a)}[\theta^{*\top}VUz] - \mathbb{E}_{z\sim P^{LD}_{t_0}(a)}[\theta^{*\top}Az] \right|}_{e_4}.$$

Then for term $e_3$, by Lemma E.2, we get
$$e_3 \leq \left| \mathbb{E}_{z\sim P^{LD}_{t_0}(a)}[\|z\|_2] - \mathbb{E}_{z\sim (U^\top V^\top)_\#\widehat{P}_a}[\|z\|_2] \right|$$
$$= \mathcal{O}\left( TV(\widehat{P}_a) \cdot \left( \sqrt{\mathbb{E}_{z\sim P^{LD}_{t_0}(a)}[\|z\|_2^2]} + \sqrt{\mathbb{E}_{x\sim \widehat{P}_a}[\|x\|_2^2]} \right) \right),$$

where we use $\mathbb{E}_{z\sim (U^\top V^\top)_\#\widehat{P}_a}[\|z\|_2] \leq \mathbb{E}_{x\sim \widehat{P}_a}[\|x\|_2]$

For $e_4$, we have
$$e_4 = \left| \mathbb{E}_{z\sim P^{LD}_{t_0}(a)}[\theta^{*\top}(VU-A)z] \right|$$
$$= \alpha(t_0)\left| \mathbb{E}_{z\sim P(a)}[\theta^{*\top}(VU-A)z] \right|$$
$$\leq \|VU - A\|_F \cdot \mathbb{E}_{z\sim P^{LD}(a)}[\|z\|_2].$$

Therefore, by combining things together, we have
$$\mathcal{E}_2 \leq e_1 + e_2 + e_3 + e_4$$
$$\leq \|VV^\top - AA^\top\|_F \cdot \sqrt{\mathbb{E}_{x\sim \widehat{P}_a}[\|x\|_2^2]} + (\|VU-A\|_F + t_0) \cdot \sqrt{M(a)}$$
$$+ \mathcal{O}\left( TV(\widehat{P}_a) \cdot \left( \sqrt{M(a) + t_0 d} + \sqrt{\mathbb{E}_{x\sim \widehat{P}_a}[\|x\|_2^2]} \right) \right).$$

By Lemma D.4 and Lemma E.1, we have
$$TV(\widehat{P}_a) = \widetilde{\mathcal{O}}\left( \sqrt{\frac{\mathcal{T}(P(x,\widehat{y}=a), P_{x\widehat{y}}; \bar{\mathcal{S}})}{\lambda_{\min}}} \cdot \epsilon_{diff} \right),$$
$$\|VV^\top - AA^\top\|_F = \widetilde{\mathcal{O}}\left( \frac{\sqrt{t_0}}{\sqrt{\lambda_{\min}}} \cdot \epsilon_{diff} \right),$$
$$\|VU - A\|_F = \mathcal{O}(d^{\frac{3}{2}}\|VV^\top - AA^\top\|_F).$$

And by Lemma E.3
$$\mathbb{E}_{x\sim \widehat{P}_a}\left[\|x\|_2^2\right] = \mathcal{O}\left( ct_0 D + M(a) \cdot (1 + TV(\widehat{P}_a)) \right).$$

Therefore. leading term in $\mathcal{E}_2$ is
$$\mathcal{E}_2 = \mathcal{O}\left( (TV(\widehat{P}_a) + t_0)\sqrt{M(a)} \right).$$

By plugging in score matching error $\epsilon_{diff}^2 = \widetilde{\mathcal{O}}\left( \frac{1}{t_0}\sqrt{\frac{Dd^2 + D^2 d}{n_1}} \right)$, we have
$$TV(\widehat{P}_a) = \widetilde{\mathcal{O}}\left( \sqrt{\frac{\mathcal{T}(P(x,\widehat{y}=a), P_{x\widehat{y}}; \bar{\mathcal{S}})}{\lambda_{\min}}} \cdot \left( \frac{Dd^2 + D^2 d}{n_1} \right)^{\frac{1}{4}} \cdot \frac{1}{\sqrt{t_0}} \right).$$

When $t_0 = \left((Dd^2 + D^2 d)/n_1\right)^{1/6}$, it admits the best trade off in $\mathcal{E}_2$ and $\mathcal{E}_2$ is bounded by

$$\mathcal{E}_2 = \widetilde{\mathcal{O}}\left(\sqrt{\frac{\mathcal{T}(P(x, \widehat{y} = a), P_{x\widehat{y}}; \bar{\mathcal{S}})}{\lambda_{\min}}} \cdot \left(\frac{Dd^2 + D^2 d}{n_1}\right)^{\frac{1}{6}} \cdot a\right).$$

$\square$

## E.6  Proof of Lemma E.1

*Proof.* From Lemma 17 in [8], we have

$$\|U - V^\top A\|_F = \mathcal{O}(\|VV^\top - AA^\top\|_F).$$

Then it suffices to bound

$$\left|\|VU - A\|_F^2 - \|U - V^\top A\|_F^2\right|,$$

where

$$\|VU - A\|_F^2 = 2d - \operatorname{trace}\left(U^\top V^\top A + A^\top VU\right)$$
$$\|U - V^\top A\|_F^2 = d + \operatorname{trace}\left(A^\top VV^\top A\right) - \operatorname{trace}\left(U^\top V^\top A + A^\top VU\right).$$

Thus

$$\left|\|VU - A\|_F^2 - \|U - V^\top A\|_F^2\right| = \left|d - \operatorname{trace}\left(A^\top VV^\top A\right)\right| = \left|\operatorname{trace}\left(AA^\top(VV^\top - AA^\top)\right)\right|,$$

which is because $\operatorname{trace}\left(A^\top VV^\top A\right)$ is calcualted as

$$\operatorname{trace}\left(A^\top VV^\top A\right) = \operatorname{trace}\left(AA^\top VV^\top\right)$$
$$= \operatorname{trace}\left(AA^\top AA^\top\right) + \operatorname{trace}\left(AA^\top(VV^\top - AA^\top)\right)$$
$$= d + \operatorname{trace}\left(AA^\top(VV^\top - AA^\top)\right).$$

Then we will bound $\left|\operatorname{trace}\left(AA^\top(VV^\top - AA^\top)\right)\right|$ by $\|VV^\top - AA^\top\|_F$,

$$\left|\operatorname{trace}\left(AA^\top(VV^\top - AA^\top)\right)\right| \leq \operatorname{trace}\left(AA^\top\right)\operatorname{trace}\left(|VV^\top - AA^\top|\right)$$
$$\leq d \cdot \operatorname{trace}\left(|VV^\top - AA^\top|\right)$$
$$\leq d \cdot \sqrt{2d \|VV^\top - AA^\top\|_F^2}.$$

Thus, $\|VU - A\|_F = \mathcal{O}\left(d^{\frac{3}{2}}\sqrt{\angle(V, A)}\right).$

$\square$

## E.7  Proof of Lemma E.2

*Proof.* When $P_1$ and $P_2$ are Gaussian, $m(z) = \|z\|_2^2$

$$\left|\mathbb{E}_{z\sim P_1}[m(z)] - \mathbb{E}_{z\sim P_2}[m(z)]\right|$$
$$= \left|\int m(z)\left(p_1(z) - p_2(z)\right)\mathrm{d}z\right|$$
$$\leq \left|\int_{\|z\|_2 \leq R} \|z\|_2^2\left(p_1(z) - p_2(z)\right)\mathrm{d}z\right| + \int_{\|z\|_2 > R} \|z\|_2^2 p_1(z)\mathrm{d}z + \int_{\|z\|_2 > R} \|z\|_2^2 p_2(z)\mathrm{d}z$$
$$\leq R^2 \mathrm{d}_{\mathrm{TV}}(P_1, P_2) + \int_{\|z\|_2 > R} \|z\|_2^2 p_1(z)\mathrm{d}z + \int_{\|z\|_2 > R} \|z\|_2^2 p_2(z)\mathrm{d}z.$$

Since $P_1$ and $P_2$ are Gaussains, $\int_{\|z\|_2 > R} \|z\|_2^2 p_1(z)\mathrm{d}z$ and $\int_{\|z\|_2 > R} \|z\|_2^2 p_2(z)\mathrm{d}z$ are bounded by some constant $C_1$ when $R^2 \geq C_2 \max\{\mathbb{E}_{z\sim P_1}[\|z\|_2^2], \mathbb{E}_{z\sim P_2}[\|z\|_2^2]\}$ as suggested by Lemma 16 in [8].

Therefore,

$$\mathbb{E}_{z \sim P_1}[\|z\|_2^2] \leq \mathbb{E}_{z \sim P_2}[\|z\|_2^2] + C_2 \max\{\mathbb{E}_{P_1}[\|z\|_2^2], \mathbb{E}_{P_2}[\|z\|_2^2]\} \cdot d_{TV}(P_1, P_2) + 2C_1$$
$$\leq \mathbb{E}_{z \sim P_2}[\|z\|_2^2] + C_2(\mathbb{E}_{z \sim P_1}[\|z\|_2^2] + \mathbb{E}_{z \sim P_2}[\|z\|_2^2]) \cdot d_{TV}(P_1, P_2) + 2C_1.$$

Then

$$\mathbb{E}_{z \sim P_1}[\|z\|_2^2] = \mathcal{O}\left(\mathbb{E}_{z \sim P_2}[\|z\|_2^2] + \mathbb{E}_{z \sim P_2}[\|z\|_2^2] \cdot d_{TV}(P_1, P_2)\right)$$

since $d_{TV}(P_1, P_2)$ decays with $n_1$.

Similarly, when $m(z) = \|z\|_2$

$$|\mathbb{E}_{z \sim P_1}[m(z)] - \mathbb{E}_{z \sim P_2}[m(z)]|$$

$$= \left| \int m(z) \, (p_1(z) - p_2(z)) \, dz \right|$$

$$\leq \left| \int_{\|z\|_2 \leq R} \|z\|_2 \, (p_1(z) - p_2(z)) \, dz \right| + \int_{\|z\|_2 > R} \|z\|_2 p_1(z) dz + \int_{\|z\|_2 > R} \|z\|_2 p_2(z) dz$$

$$\leq R d_{TV}(P_1, P_2) + \sqrt{\int_{\|z\|_2 > R} \|z\|_2^2 p_1(z) dz} + \sqrt{\int_{\|z\|_2 > R} \|z\|_2^2 p_2(z) dz},$$

where $\int_{\|z\|_2 > R} \|z\|_2^2 p_1(z) dz$ and $\int_{\|z\|_2 > R} \|z\|_2^2 p_2(z) dz$ are bounded by some constant $C_1$ when $R^2 \geq C_2 \max\{\mathbb{E}_{z \sim P_1}[\|z\|_2^2], \mathbb{E}_{z \sim P_2}[\|z\|_2^2]\}$ as suggested by Lemma 16 in [8].

Therefore,

$$|\mathbb{E}_{z \sim P_1}[\|z\|_2] - \mathbb{E}_{z \sim P_2}[\|z\|_2]| \leq \sqrt{C_2 \max\{\mathbb{E}_{P_1}[\|z\|_2^2], \mathbb{E}_{P_2}[\|z\|_2^2]\} \cdot d_{TV}(P_1, P_2)} + 2C_1$$

$$\leq \left(\sqrt{C_2 \mathbb{E}_{z \sim P_1}[\|z\|_2^2]} + \sqrt{C_2 \mathbb{E}_{z \sim P_2}[\|z\|_2^2]}\right) \cdot d_{TV}(P_1, P_2) + 2C_1$$

$$= \mathcal{O}\left(\left(\sqrt{\mathbb{E}_{z \sim P_1}[\|z\|_2^2]} + \sqrt{\mathbb{E}_{z \sim P_2}[\|z\|_2^2]}\right) \cdot d_{TV}(P_1, P_2)\right).$$

$\square$

### E.8 Proof of Lemma E.3

*Proof.* Recall from (E.12) that

$$P^{LD}(a) = P_z(z \mid \widehat{f}(Az) = a) = \mathsf{N}\left(\mu(a), \Gamma\right)$$

with $\mu(a) := \Sigma\widehat{\beta}\left(\widehat{\beta}^\top \Sigma \widehat{\beta} + \nu^2\right)^{-1} a$, $\Gamma := \Sigma - \Sigma\widehat{\beta}\left(\widehat{\beta}^\top \Sigma \widehat{\beta} + \nu^2\right)^{-1}\widehat{\beta}^\top \Sigma$.

$$\mathbb{E}_{z \sim P^{LD}(a)}\left[\|z\|_2^2\right] = \mu(a)^\top \mu(a) + \text{trace}(\Gamma)$$

$$= \frac{\widehat{\beta}^\top \Sigma^2 \widehat{\beta}}{\left(\|\widehat{\beta}\|_\Sigma^2 + \nu^2\right)^2} a^2 + \text{trace}(\Sigma - \Sigma\widehat{\beta}\left(\widehat{\beta}^\top \Sigma \widehat{\beta} + \nu^2\right)^{-1}\widehat{\beta}^\top \Sigma)$$

$$=: M(a).$$

$$M(a) = \mathcal{O}\left(\frac{\widehat{\beta}^\top \Sigma^2 \widehat{\beta}}{\left(\|\widehat{\beta}\|_\Sigma^2\right)^2} a^2 + \text{trace}(\Sigma)\right)$$

$$= \mathcal{O}\left(\frac{a^2}{\|\widehat{\beta}\|_\Sigma} + d\right),$$

and by Lemma D.6

$$\|\widehat{\beta}\|_\Sigma \leq \frac{1}{2}\|\beta^*\|_\Sigma.$$

Thus $\mathbb{E}_{z\sim PLD(a)}\left[\|z\|_2^2\right] = M(a), M(a) = \mathcal{O}\left(\frac{a^2}{\|\beta^*\|_\Sigma} + d\right).$

Thus after adding diffusion noise at $t_0$, we have for $\alpha(t) = e^{-t/2}$ and $h(t) = 1 - e^{-t}$:

$$
\begin{aligned}
\mathbb{E}_{z\sim P_{t_0}^{LD}(a)}\left[\|z\|_2^2\right] &= \mathbb{E}_{z_0\sim PLD(a)}\mathbb{E}_{z\sim \mathsf{N}(\alpha(t_0)\cdot z_0, h(t_0)\cdot \mathsf{Id})}\left[\|z\|_2^2\right] \\
&= \mathbb{E}_{z_0\sim PLD(a)}\left[\alpha^2(t_0)\|z_0\|_2^2 + d\cdot h(t_0)\right] \\
&= \alpha^2(t_0)\cdot \mathbb{E}_{z_0\sim PLD(a)}\left[\|z_0\|_2^2\right] + d\cdot h(t_0) \\
&= e^{-t_0}\cdot \mathbb{E}_{z_0\sim PLD(a)}\left[\|z_0\|_2^2\right] + (1 - e^{-t_0})\cdot d.
\end{aligned}
$$

Thus $\mathbb{E}_{z\sim P_{t_0}^{LD}(a)}\left[\|z\|_2^2\right] \leq M(a) + t_0 d.$

By orthogonal decomposition we have

$$
\begin{aligned}
\mathbb{E}_{x\sim \widehat{P}_a}\left[\|x\|_2^2\right] &\leq \mathbb{E}_{x\sim \widehat{P}_a}\left[\|(I_D - VV^\top)x\|_2^2\right] + \mathbb{E}_{x\sim \widehat{P}_a}\left[\|VV^\top x\|_2^2\right] \\
&= \mathbb{E}_{x\sim \widehat{P}_a}\left[\|(I_D - VV^\top)x\|_2^2\right] + \mathbb{E}_{x\sim \widehat{P}_a}\left[\|U^\top V^\top x\|_2^2\right],
\end{aligned}
$$

where $\mathbb{E}_{x\sim \widehat{P}_a}\left[\|(I_D - VV^\top)x\|_2^2\right]$ is bounded by (D.5) and the distribution of $U^\top V^\top x$, which is $(U^\top V^\top)_\# \widehat{P}_a$, is close to $\mathbb{P}_{t_0}^{LD}(a)$ up to $TV(\widehat{P}_a)$, which is defined in Definition D.2. Then by Lemma E.2, we have

$$\mathbb{E}_{x\sim \widehat{P}_a}\left[\|U^\top V^\top x\|_2^2\right] = \mathcal{O}\left(\mathbb{E}_{z\sim P_{t_0}^{LD}(a)}\left[\|z\|_2^2\right](1 + TV(\widehat{P}_a))\right).$$

Thus $\mathbb{E}_{x\sim \widehat{P}_a}\left[\|x\|_2^2\right] = \mathcal{O}\left(ct_0 D + (M(a) + t_0 d)\cdot(1 + TV(\widehat{P}_a))\right).$

$\square$

## E.9   Proof of Lemma E.4

*Proof.* Firstly, one can verify the following two equations by direct calculation:

$$
\begin{aligned}
(\lambda I_D + A\Sigma_1 A^\top)^{-1} &= \frac{1}{\lambda}\left(I_D - A(\lambda I_d + \Sigma_1)^{-1}\Sigma_1 A^\top\right), \\
(\lambda I_d + \Sigma_1)^{-1} &= \frac{1}{\lambda}\left(I_d - (\lambda I_d + \Sigma_1)^{-1}\Sigma_1\right).
\end{aligned}
$$

Then we have

$$
\begin{aligned}
(\lambda I_D + A\Sigma_1 A^\top)^{-1}A\Sigma_2 A^\top &= \frac{1}{\lambda}\left(I_D - A(\lambda I_d + \Sigma_1)^{-1}\Sigma_1 A^\top\right)A\Sigma_2 A^\top \\
&= \frac{1}{\lambda}\left(A\Sigma_2 A^\top - A(\lambda I_d + \Sigma_1)^{-1}\Sigma_1 \Sigma_2 A^\top\right).
\end{aligned}
$$

Therefore,

$$
\begin{aligned}
\mathrm{Tr}\left((\lambda I_D + A\Sigma_1 A^\top)^{-1}A\Sigma_2 A^\top\right) &= \mathrm{Tr}\left(\frac{1}{\lambda}\left(A\Sigma_2 A^\top - A(\lambda I_d + \Sigma_1)^{-1}\Sigma_1 \Sigma_2 A^\top\right)\right) \\
&= \mathrm{Tr}\left(\frac{1}{\lambda}\left(\Sigma_2 - (\lambda I_d + \Sigma_1)^{-1}\Sigma_1 \Sigma_2\right)\right) \\
&= \mathrm{Tr}\left(\frac{1}{\lambda}\left(I_d - (\lambda I_d + \Sigma_1)^{-1}\Sigma_1\right)\Sigma_2\right) \\
&= \mathrm{Tr}\left((\lambda I_d + \Sigma_1)^{-1}\Sigma_2\right),
\end{aligned}
$$

which has finished the proof.

$\square$

# F Theory in Nonparametric Setting

Built upon the insights from Section 4, we provide analysis to the nonparametric reward and general data sampling setting. We generalize Assumption 4.2 to the following.

**Assumption F.1** . The ground truth reward $f^*$ is decomposed as

$$f^*(x) = g^*(x_\parallel) - h^*(x_\perp),$$

where $g^*(x_\parallel)$ is $\alpha$-Hölder continuous for $\alpha \geq 1$ and $h^*(x_\perp)$ is nondecreasing in terms of $\|x_\perp\|_2$ with $h^*(0) = 0$. Moreover, $g^*$ has a bounded Hölder norm, i.e., $\|g^*\|_{\mathcal{H}^\alpha} \leq 1$.

Hölder continuity is widely studied in nonparametric statistics literature [15, 49]. $h^*$ here penalizes off-support extrapolation.

Under Assumption F.1, we use nonparametric regression for estimating $f^*$. Specifically, we specialize (3.1) in Algorithm 1 to

$$\widehat{f_\theta} \in \operatorname*{argmin}_{f_\theta \in \mathcal{F}} \frac{1}{2n} \sum_{i=1}^{n_1} (f_\theta(x_i) - y_i)^2,$$

where $\mathcal{F} = \mathrm{NN}(L, M, J, K, \kappa)$ is chosen to be a class of neural networks. Hyperparameters in $\mathcal{F}$ will be chosen properly in Theorem F.4.

Our theory also considers generic sampling distributions on $x$. Since $x$ lies in a low-dimensional subspace, this translates to a sampling distribution assumption on latent variable $z$.

**Assumption F.2** . The latent variable $z$ follows distribution $P_z$ with density $p_z$, such that there exists constants $B, C_1, C_2$ verifying $p_z(z) \leq (2\pi)^{-(d+1)/2} C_1 \exp\left(-C_2 \|z\|_2^2/2\right)$ whenever $\|z\|_2 > B$. And $c_0 I_d \preceq \mathbb{E}_{z \sim P_z}\left[zz^\top\right]$.

Assumption F.2 says $P_z$ has a light tail, which is standard in high-dimensional statistics [51, 53]. Assumption F.2 also encodes distributions with a compact support. Furthermore, we assume that the curated data $(x, \widehat{y})$ induces Lipschitz conditional scores. Motivated by Chen et al. [8], we show that the linear subspace structure in $x$ leads to a similar conditional score decomposition $\nabla \log p_t(x|\widehat{y}) = s_\parallel(x, \widehat{y}, t) + s_\perp(x, \widehat{y}, t)$, where $s_\parallel$ is the on-support score and $s_\perp$ is the orthogonal score. The decomposition for conditional score is as (H.1), which applies to both parametric and non-parametric cases. The following assumption is imposed on $s_\parallel$.

**Assumption F.3** . The on-support conditional score function $s_\parallel(x, \widehat{y}, t)$ is Lipschitz with respect to $x, \widehat{y}$ for any $t \in (0, T]$, i.e., there exists a constant $C_{\mathrm{lip}}$, such that for any $x, \widehat{y}$ and $x', \widehat{y}'$, it holds

$$\|s_\parallel(x, \widehat{y}, t) - s_\parallel(x', \widehat{y}', t)\|_2 \leq C_{\mathrm{lip}}\|x - x'\|_2 + C_{\mathrm{lip}}|\widehat{y} - \widehat{y}'|_2.$$

Lipschitz score is commonly adopted in existing works [9, 25]. Yet Assumption F.3 only requires the Lipschitz continuity of the on-support score, which matches the weak regularity conditions in Lee et al. [25], Chen et al. [8]. We then choose the score network architecture similar to that in the linear reward setting, except we replace $m$ by a nonlinear network. Recall the linear encoder and decoder estimate the representation matrix $A$.

We consider feedforward networks with ReLU activation functions as concept classes $\mathcal{F}$ and $\mathcal{S}$ for nonparametric regression and conditional score matching. Generalization to different network architectures poses no real difficulty. Given an input $x$, neural networks compute

$$f_{\mathrm{NN}}(x) = W_L \sigma(\ldots \sigma(W_1 x + b_1)\ldots) + b_L, \tag{F.1}$$

where $W_i$ and $b_i$ are weight matrices and intercepts, respectively. We then define a class of neural networks as

$$\mathrm{NN}(L, M, J, K, \kappa) = \Big\{ f : f \text{ in the form of (F.1) with } L \text{ layers and width bounded by } M,$$

$$\sup_x \|f(x)\|_2 \leq K, \max\{\|b_i\|_\infty, \|W_i\|_\infty\} \leq \kappa \text{ for } i = 1, \ldots, L, \text{ and } \sum_{i=1}^L \left(\|W_i\|_0 + \|b_i\|_0\right) \leq J\Big\}.$$

For the conditional score network, we will additionally impose some Lipschitz continuity requirement, i.e., $\|f(x) - f(y)\|_2 \leq c_{\text{lip}}\|x - y\|_2$ for some Lipschitz coefficient $c_{\text{lip}}$.

Recall the distribution shift defined in Definition D.3 that

$$\mathcal{T}(P_1, P_2; \mathcal{L}) = \sup_{l \in \mathcal{L}} \mathbb{E}_{x \sim P_1}[l(x)]/\mathbb{E}_{x \sim P_2}[l(x)]$$

for arbitrary two distributions $P_1, P_2$ and function class $\mathcal{L}$. Similar to the parametric case, use notation $\widehat{P}_a := \widehat{P}(\cdot|\widehat{y} = a)$ and $P_a := P(\cdot|\widehat{y} = a)$. Then we can bound $\texttt{SubOpt}(\widehat{P}_a; y^* = a)$ in Theorem F.4 in terms of non-parametric regression error, score matching error and distribution shifts in both regression and score matching.

**Theorem F.4.** Suppose Assumption 4.1, F.1, F.2 and F.3 hold. Let $\delta(n) = \frac{d \log \log n}{\log n}$. Properly chosen $\mathcal{F}$ and $\mathcal{S}$, with high probability, running Algorithm 1 with a target reward value $a$ and stopping at $t_0 = \left( n_1^{-\frac{2-2\delta(n_1)}{d+6}} + Dn_1^{-\frac{d+4}{d+6}} \right)^{\frac{1}{3}}$ gives rise to $\angle(V, A) \leq \widetilde{\mathcal{O}}\left( \frac{1}{c_0}\left( n_1^{-\frac{2-2\delta(n_1)}{d+6}} + Dn_1^{-\frac{d+4}{d+6}} \right) \right)$ and

$$
\begin{aligned}
&\texttt{SubOpt}(\widehat{P}_a; y^* = a) \\
&\leq \underbrace{\sqrt{\mathcal{T}(P(x|\widehat{y} = a), P_x; \bar{\mathcal{F}})} \cdot \widetilde{\mathcal{O}}\left( n_2^{-\frac{\alpha - \delta(n_2)}{2\alpha + d}} + D/n_2 \right)}_{\mathcal{E}_1} \\
&+ \underbrace{\left( \sqrt{\frac{\mathcal{T}(P(x, \widehat{y} = a), P_{x\widehat{y}}; \bar{\mathcal{S}})}{c_0}} \cdot \|g^*\|_\infty + \sqrt{M(a)} \right) \cdot \widetilde{\mathcal{O}}\left( \left( n_1^{-\frac{2-2\delta(n_1)}{d+6}} + Dn_1^{-\frac{d+4}{d+6}} \right)^{\frac{1}{3}} \right)}_{\mathcal{E}_2} \\
&+ \underbrace{\mathbb{E}_{x \sim \widehat{P}_a}[h^*(x_\perp)]}_{\mathcal{E}_3},
\end{aligned}
$$

where $M(a) := \mathbb{E}_{z \sim \mathbb{P}(a)}[\|z\|_2^2]$ and

$$\bar{\mathcal{F}} := \{|f^*(x) - f(x)|^2 : f \in \mathcal{F}\}, \quad \bar{\mathcal{S}} = \left\{ \frac{1}{T - t_0}\int_{t_0}^{T} \mathbb{E}_{x_t|x}\|\nabla \log p_t(x_t \mid y) - s(x_t, y, t)\|_2^2 \mathrm{d}t : s \in \mathcal{S} \right\},$$

$\mathcal{E}_3$ penalizes the component in $\widehat{P}_a$ that is off the truth subspace. The function classes $\mathcal{F}$ and $\mathcal{S}$ are chosen as $\mathcal{F} = \text{NN}(L_f, M_f, J_f, K_f, \kappa_f)$ with

$$L_f = \mathcal{O}(\log n_2), \ M_f = \mathcal{O}\left( n_2^{-\frac{d}{d+2\alpha}}(\log n_2)^{d/2} \right), \ J_f = \mathcal{O}\left( n_2^{-\frac{d}{d+2\alpha}}(\log n_2)^{d/2+1} \right)$$

$$K_f = 1, \ \kappa_f = \mathcal{O}\left( \sqrt{\log n_2} \right)$$

and $\mathcal{S} = \text{NN}(L_s, M_s, J_s, K_s, \kappa_s)$ with

$$L_s = \mathcal{O}(\log n_1 + d), \ M_s = \mathcal{O}\left( d^{d/2}n_1^{-\frac{d+2}{d+6}}(\log n_1)^{d/2} \right), \ J_s = \mathcal{O}\left( d^{d/2}n_1^{-\frac{d+2}{d+6}}(\log n_1)^{d/2+1} \right)$$

$$K_s = \mathcal{O}\left( d\log(dn_1) \right), \ \kappa_s = \mathcal{O}\left( \sqrt{d\log(n_1 d)} \right).$$

Moreover, $\mathcal{S}$ is also Lipschitz with respect to $(x, y)$ and the Lipschitz coefficient is $c_{\text{lip}} = \mathcal{O}(10dC_{\text{lip}})$.

**Remark.** The proof is provided in Appendix G.2. Here we correct a typo on $n_1^{-\frac{1}{3(d+5)}}$ in main paper Section 4.3. Quantities $\mathcal{T}(P(x|\widehat{y} = a), P_x; \bar{\mathcal{F}})$ and $\mathcal{T}(P(x, \widehat{y} = a), P_{x\widehat{y}}; \bar{\mathcal{S}})$ depend on $a$ characterizing the distribution shift. The $\delta(n)$ terms account for the unbounded domain of $x$, which is negligible when $n$ is large. In the main paper, we omit $\delta(n)$ in the regret bound.

# G  Omitted Proofs in Section F

## G.1  Conditional Score Decomposition and Score Matching Error

**Lemma G.1.** Under Assumption 4.1, F.2 and F.3, with high probability

$$\frac{1}{T-t_0} \int_{t_0}^{T} \|\widehat{s}(\cdot, t) - \nabla \log p_t(\cdot)\|_{L^2(P_t)}^2 \mathrm{d}t \le \epsilon_{diff}^2(n_1)$$

with $\epsilon_{diff}^2(n_1) = \widetilde{\mathcal{O}}\left(\frac{1}{t_0}\left(n_1^{-\frac{2-2\delta(n_1)}{d+6}} + Dn_1^{-\frac{d+4}{d+6}}\right)\right)$ for $\delta(n_1) = \frac{d \log \log n_1}{\log n_1}$.

*Proof.* [8, Theorem 1] is easily adapted here to prove Lemma G.1 with the input dimension $d+1$ and the Lipschitzness in Assumption F.3. Network size of $\mathcal{S}$ is implied by [8, Theorem 1] with $\epsilon = n_1^{-\frac{1}{d+6}}$ accounting for the additional dimension of reward $\widehat{y}$ and then the score matching error follows.  □

## G.2  Proof of Theorem F.4

**Additional Notations:**  Similar as before, use $P_t^{LD}(z)$ to denote the low-dimensional distribution on $z$ corrupted by diffusion noise. Formally, $p_t^{LD}(z) = \int \phi_t(z'|z)p_z(z)\mathrm{d}z$ with $\phi_t(\cdot|z)$ being the density of $\mathsf{N}(\alpha(t)z, h(t)I_d)$. $P_{t_0}^{LD}(z \mid \widehat{f}(Az) = a)$ the corresponding conditional distribution on $\widehat{f}(Az) = a$ at $t_0$, with shorthand as $P_{t_0}^{LD}(a)$. Also give $P_z(z \mid \widehat{f}(Az) = a)$ a shorthand as $P^{LD}(a)$.

### G.2.1  $\mathtt{SubOpt}(\widehat{P}_a; y^* = a)$ **Decomposition**

By the same argument as in §D.3.1, we have

$$\mathtt{SubOpt}(\widehat{P}_a; y^* = a) \le \underbrace{\mathbb{E}_{x \sim P_a}\left[\left|f^*(x) - \widehat{f}(x)\right|\right]}_{\mathcal{E}_1} + \underbrace{\left|\mathbb{E}_{x \sim P_a}[g^*(x_\|)] - \mathbb{E}_{x \sim \widehat{P}_a}[g^*(x_\|)]\right|}_{\mathcal{E}_2}$$

$$+ \underbrace{\mathbb{E}_{x \sim \widehat{P}_a}[h^*(x_\perp)]}_{\mathcal{E}_3}.$$

### G.2.2  $\mathcal{E}_1$: **Nonparamtric Regression Induced Error**

**Nonparametric Regression Error of** $\widehat{f}$  Since $P_z$ has a light tail due to Assumption F.2, by union bound and [8, Lemma 16], we have

$$\mathbb{P}(\exists\, x_i \text{ with } \|x_i\|_2 > R \text{ for } i = 1, \ldots, n_2) \le n_2 \frac{C_1 d2^{-d/2+1}}{C_2 \Gamma(d/2+1)} R^{d-2} \exp(-C_2 R^2/2),$$

where $C_1, C_2$ are constants and $\Gamma(\cdot)$ is the Gamma function. Choosing $R = \mathcal{O}(\sqrt{d \log d + \log \frac{n}{\delta}})$ ensures $\mathbb{P}(\exists\, x_i \text{ with } \|x_i\|_2 > R \text{ for } i = 1, \ldots, n_2) < \delta$. On the event $\mathcal{E} = \{\|x_i\|_2 \le R \text{ for all } i = 1, \ldots, n_2\}$, denoting $\delta(n_2) = \frac{d \log \log n_2}{\log n_2}$, we have

$$\|f^* - \widehat{f}\|_{L^2}^2 = \widetilde{\mathcal{O}}\left(n_2^{-\frac{2(\alpha-\delta(n_2))}{d+2\alpha}}\right)$$

by [31, Theorem 7] with a new covering number of $\mathcal{S}$, when $n_2$ is sufficiently large. The corresponding network architecture follows from Theorem 2 in "Nonparametric Regression on Low-Dimensional Manifolds using Deep ReLU Networks : Function Approximation and Statistical Recovery".

We remark that linear subspace is a special case of low Minkowski dimension. Moreover, $\delta(n_2)$ is asymptotically negligible and accounts for the truncation radius $R$ of $x_i$'s (see also [8, Theorem 2 and 3]). The covering number of $\mathcal{S}$ is $\widetilde{\mathcal{O}}\left(d^{d/2}n^{-\frac{d}{\alpha}}(\log n_2)^{d/2} + Dd\right)$ as appear in [8, Proof of Theorem

2]. Therefore

$$\mathbb{E}_{x \sim P_a}\left[\left|f^*(x) - \widehat{f}(x)\right|\right] \le \sqrt{\mathbb{E}_{x \sim P_a}\left[\left|f^*(x) - \widehat{f}(x)\right|^2\right]}$$

$$\le \sqrt{\mathcal{T}(P(x|\widehat{y} = a), P_x; \bar{\mathcal{F}}) \cdot \|f^* - \widehat{f}\|_{L^2}^2}$$

$$= \sqrt{\mathcal{T}(P(x|\widehat{y} = a), P_x; \bar{\mathcal{F}})} \cdot \widetilde{\mathcal{O}}\left(n_2^{-\frac{\alpha - \delta(n_2)}{2\alpha + d}} + D/n_2\right).$$

### G.2.3 $\mathcal{E}_2$: Diffusion Induced On-support Error

Suppose $L_2$ score matching error is $\epsilon_{diff}^2(n_1)$, i.e.

$$\frac{1}{T - t_0}\int_{t_0}^{T} \mathbb{E}_{x,\widehat{f}}\|\nabla_x \log p_t(x, \widehat{f}) - s_{\widehat{w}}(x, \widehat{f}, t)\|_2^2 \mathrm{d}t \le \epsilon_{diff}^2(n_1),$$

We revoke Definition D.2 measuring the distance between $\widehat{P}_a$ to $P_a$ that

$$TV(\widehat{P}_a) := \mathsf{d}_{\mathrm{TV}}\left(P_{t_0}^{LD}(z \mid \widehat{f}(Az) = a), (U^\top V^\top)_\# \widehat{P}_a\right).$$

Lemma D.4 applies to nonparametric setting, so we have

$$(I_D - VV^\top)x \sim \mathsf{N}(0, \Lambda), \quad \Lambda \prec ct_0 I_D, \tag{G.1}$$

$$\angle(V, A) = \widetilde{\mathcal{O}}\left(\frac{t_0}{c_0} \cdot \epsilon_{diff}^2(n_1)\right). \tag{G.2}$$

In addition,

$$TV(\widehat{P}_a) = \widetilde{\mathcal{O}}\left(\sqrt{\frac{\mathcal{T}(P(x, \widehat{y} = a), P_{x\widehat{y}}; \bar{\mathcal{S}})}{c_0}} \cdot \epsilon_{diff}(n_1)\right). \tag{G.3}$$

$\mathcal{E}_2$ will be bounded by

$$\mathcal{E}_2 = \left|\mathbb{E}_{x \sim P_a}[g^*(x)] - \mathbb{E}_{x \sim \widehat{P}_a}[g^*(x)]\right|$$

$$\le \left|\mathbb{E}_{x \sim P_a}[g^*(AA^\top x)] - \mathbb{E}_{x \sim \widehat{P}_a}[g^*(VV^\top x)]\right| + \left|\mathbb{E}_{x \sim \widehat{P}_a}[g^*(VV^\top x) - g^*(AA^\top x)]\right|,$$

where for $\left|\mathbb{E}_{x \sim \widehat{P}_a}[g^*(VV^\top x) - g^*(AA^\top x)]\right|$, we have

$$\left|\mathbb{E}_{x \sim \widehat{P}_a}[g^*(VV^\top x) - g^*(AA^\top x)]\right| \le \mathbb{E}_{x \sim \widehat{P}_a}[\|VV^\top x - AA^\top x\|_2] \le \|VV^\top - AA^\top\|_F \cdot \mathbb{E}_{x \sim \widehat{P}_a}[\|x\|_2].$$
$$\tag{G.4}$$

For the other term $\left|\mathbb{E}_{x \sim P_a}[g^*(AA^\top x)] - \mathbb{E}_{x \sim \widehat{P}_a}[g^*(VV^\top x)]\right|$, we will bound it with $TV(\widehat{P}_a)$.

$$\left|\mathbb{E}_{x \sim P_a}[g^*(AA^\top x)] - \mathbb{E}_{x \sim \widehat{P}_a}[g^*(VV^\top x)]\right|$$

$$\le \left|\mathbb{E}_{z \sim \mathbb{P}_{t_0}(a)}[g^*(Az)] - \mathbb{E}_{z \sim (V^\top)_\# \widehat{P}_a}[g^*(Vz)]\right| + \left|\mathbb{E}_{z \sim \mathbb{P}(a)}[g^*(Az)] - \mathbb{E}_{z \sim \mathbb{P}_{t_0}(a)}[g^*(Az)]\right|$$

Since any $z \sim \mathbb{P}_{t_0}(a)$ can be represented by $\alpha(t_0)z + \sqrt{h(t_0)}u$, where $z \sim \mathbb{P}(a), u \sim \mathsf{N}(0, I_d)$, then

$$\mathbb{E}_{z \sim \mathbb{P}_{t_0}(a)}[g^*(Az)]$$

$$= \mathbb{E}_{z \sim \mathbb{P}(a), u \sim \mathsf{N}(0, I_d)}[g^*(\alpha(t)Az + \sqrt{h(t)}Au))]$$

$$\le \mathbb{E}_{z \sim \mathbb{P}(a)}[g^*(\alpha(t_0)Az))] + \sqrt{h(t_0)}\mathbb{E}_{u \sim \mathsf{N}(0, I_d)}[\|Au\|_2]$$

$$\le \mathbb{E}_{z \sim \mathbb{P}(a)}[g^*(Az))] + (1 - \alpha(t_0))\mathbb{E}_{z \sim \mathbb{P}(a)}[\|Az\|_2] + \sqrt{h(t_0)}\mathbb{E}_{u \sim \mathsf{N}(0, I_d)}[\|Au\|_2],$$

thus
$$\left| \mathbb{E}_{z \sim \mathbb{P}(a)}[g^*(Az)] - \mathbb{E}_{z \sim \mathbb{P}_{t_0}(a)}[g^*(Az)] \right| \le t_0 \cdot \mathbb{E}_{z \sim \mathbb{P}(a)}[\|z\|_2] + d,$$

where we further use $1 - \alpha(t_0) = 1 - e^{-t_0/2} \le t_0/2$, $h(t_0) \le 1$.

As for $\left| \mathbb{E}_{z \sim \mathbb{P}_{t_0}(a)}[g^*(Az)] - \mathbb{E}_{z \sim (V^\top)_\# \widehat{P}_a}[g^*(Vz)] \right|$, we have

$$\left| \mathbb{E}_{z \sim \mathbb{P}_{t_0}(a)}[g^*(Az)] - \mathbb{E}_{z \sim (U^\top V^\top)_\# \widehat{P}_a}[g^*(VUz)] \right|$$
$$= \left| \mathbb{E}_{z \sim \mathbb{P}_{t_0}(a)}[g^*(VUz)] - \mathbb{E}_{z \sim (U^\top V^\top)_\# \widehat{P}_a}[g^*(VUz)] \right| + \left| \mathbb{E}_{z \sim \mathbb{P}_{t_0}(a)}[g^*(Az)] - \mathbb{E}_{z \sim \mathbb{P}_{t_0}(a)}[g^*(VUz)] \right|,$$

where

$$\left| \mathbb{E}_{z \sim \mathbb{P}_{t_0}(a)}[g^*(Az)] - \mathbb{E}_{z \sim \mathbb{P}_{t_0}(a)}[g^*(VUz)] \right| \le \|A - VU\|_F \cdot \mathbb{E}_{z \sim \mathbb{P}_{t_0}(a)}[\|z\|_2],$$

and

$$\left| \mathbb{E}_{z \sim \mathbb{P}_{t_0}(a)}[g^*(VUz)] - \mathbb{E}_{z \sim (U^\top V^\top)_\# \widehat{P}_a}[g^*(VUz)] \right| \le TV(\widehat{P}_a) \cdot \|g^*\|_\infty.$$

Combining things up, we have

$$\begin{aligned}
\mathcal{E}_2 \le & \|VV^\top - AA^\top\|_F \cdot \mathbb{E}_{x \sim \widehat{P}_a}[\|x\|_2] + \|A - VU\|_F \cdot \mathbb{E}_{z \sim \mathbb{P}_{t_0}(a)}[\|z\|_2] \\
& + t_0 \cdot \mathbb{E}_{z \sim \mathbb{P}(a)}[\|z\|_2] + d + TV(\widehat{P}_a) \cdot \|g^*\|_\infty.
\end{aligned}$$

Similar to parametric case, Let $M(a) := \mathbb{E}_{z \sim \mathbb{P}(a)}[\|z\|_2^2]$, then

$$\mathbb{E}_{z \sim \mathbb{P}_{t_0}(a)}[\|z\|_2^2] \le M(a) + t_0 d,$$

expect for in nonparametric case, we can not compute $M(a)$ out as it is not Gaussian. But still, with higher-order terms in $n_1^{-1}$ hided, we have

$$\begin{aligned}
\mathcal{E}_2 &= \mathcal{O}\left( TV(\widehat{P}_a) \cdot \|g^*\|_\infty + t_0 M(a) \right) \\
&= \widetilde{\mathcal{O}}\left( \sqrt{\frac{\mathcal{T}(P(x, \widehat{y} = a), P_{x\widehat{y}}; \bar{\mathcal{S}})}{c_0}} \cdot \epsilon_{diff}(n_1) \cdot \|g^*\|_\infty + t_0 M(a) \right).
\end{aligned}$$

## H   Parametric Conditional Score Estimation: Proof of Lemma D.1

*Proof.* We first derive a decomposition of the conditional score function similar to [8]. We have

$$\begin{aligned}
p_t(x, y) &= \int p_t(x, y | z) p_z(z) \mathrm{d}z \\
&= \int p_t(x | z) p(y | z) p_z(z) \mathrm{d}z \\
&= C \int \exp\left( -\frac{1}{2h(t)} \|x - \alpha(t) Az\|_2^2 \right) \exp\left( -\frac{1}{\sigma_y^2} \left( \theta^\top z - y \right)^2 \right) p_z(z) \mathrm{d}z \\
&\stackrel{(i)}{=} C \exp\left( -\frac{1}{2h(t)} \|(I_D - AA^\top) x\|_2^2 \right) \\
&\quad \cdot \int \exp\left( -\frac{1}{2h(t)} \|A^\top x - \alpha(t) z\|_2^2 \right) \exp\left( -\frac{1}{\sigma_y^2} \left( \theta^\top z - y \right)^2 \right) p_z(z) \mathrm{d}z,
\end{aligned}$$

where equality $(i)$ follows from the fact $AA^\top x \perp (I_D - AA^\top) x$ and $C$ is the normalizing constant of Gaussian densities. Taking logarithm and then derivative with respect to $x$ on $p_t(x, y)$, we obtain

$$\begin{aligned}
& \nabla_x \log p_t(x, y) \\
&= \frac{\alpha(t)}{h(t)} \frac{A \int z \exp\left( -\frac{1}{2h(t)} \|A^\top x - \alpha(t) z\|_2^2 \right) \exp\left( -\frac{1}{\sigma_y^2} \left( \theta^\top z - y \right)^2 \right) p_z(z) \mathrm{d}z}{\int \exp\left( -\frac{1}{2h(t)} \|A^\top x - \alpha(t) z\|_2^2 \right) \exp\left( -\frac{1}{\sigma_y^2} \left( \theta^\top z - y \right)^2 \right) p_z(z) \mathrm{d}z} - \frac{1}{h(t)} x.
\end{aligned}$$

Note that the first term in the right-hand side above only depends on $A^\top x$ and $y$. Therefore, we can compactly write $\nabla_x \log p_t(x, y)$ as

$$\nabla_x \log p_t(x, y) = \frac{1}{h(t)} A u(A^\top x, y, t) - \frac{1}{h(t)} x, \tag{H.1}$$

where mapping $u$ represents

$$\frac{\alpha(t) \int z \exp\left(-\frac{1}{2h(t)} \|A^\top x - \alpha(t) z\|_2^2\right) \exp\left(-\frac{1}{\sigma_y^2}\left(\theta^\top z - y\right)^2\right) p_z(z) \mathrm{d}z}{\int \exp\left(-\frac{1}{2h(t)} \|A^\top x - \alpha(t) z\|_2^2\right) \exp\left(-\frac{1}{\sigma_y^2}\left(\theta^\top z - y\right)^2\right) p_z(z) \mathrm{d}z}.$$

We observe that (H.1) motivates our choice of the neural network architecture $\mathcal{S}$ in (3.8). In particular, $\psi$ attempts to estimate $u$ and matrix $V$ attempts to estimate $A$.

In the Gaussian design case (Assumption 4.4), we instantiate $p_z(z)$ to the Gaussian density $(2\pi|\Sigma|)^{-d/2} \exp\left(-\frac{1}{2} z^\top \Sigma^{-1} z\right)$. Some algebra on the Gaussian integral gives rise to

$$\nabla_x \log p_t(x, y) = \frac{\alpha(t)}{h(t)} A B_t \mu_t(x, y) - \frac{1}{h(t)}(I_D - AA^\top)x - \frac{1}{h(t)} AA^\top x$$

$$= \frac{\alpha(t)}{h(t)} A B_t \left(\alpha(t) A^\top x + \frac{h(t)}{\nu^2} y\theta\right) - \frac{1}{h(t)} x, \tag{H.2}$$

where we have denoted

$$\mu_t(x, y) = \alpha(t) A^\top x + \frac{h(t)}{\nu^2} y\theta \quad \text{and} \quad B_t = \left(\alpha^2(t) I_d + \frac{h(t)}{\nu^2}\theta\theta^\top + h(t)\Sigma^{-1}\right)^{-1}.$$

**Score Estimation Error**   Recall that we estimate the conditional score function via minimizing the denoising score matching loss in Proposition 3.1. To ease the presentation, we denote

$$\ell(x, y; s) = \frac{1}{T - t_0} \int_{t_0}^{T} \mathbb{E}_{x'|x} \|\nabla_{x'} \log \phi_t(x'|x) - s(x', y, t)\|_2^2 \mathrm{d}t$$

as the loss function for a pair of clean data $(x, y)$ and a conditional score function $s$. Further, we denote the population loss as

$$\mathcal{L}(s) = \mathbb{E}_{x,y}[\ell(x, y; s)],$$

whose empirical counterpart is denoted as $\widehat{\mathcal{L}}(s) = \frac{1}{n_1} \sum_{i=1}^{n_1} \ell(x_i, y_i; s)$.

To bound the score estimation error, we begin with an oracle inequality. Denote $\mathcal{L}^{\mathrm{trunc}}(s)$ as a truncated loss function defined as

$$\mathcal{L}^{\mathrm{trunc}}(s) = \mathbb{E}[\ell(x, y; s)\mathbb{1}\{\|x\|_2 \le R, |y| \le R\}],$$

where $R > 0$ is a truncation radius chosen as $\mathcal{O}(\sqrt{d \log d + \log K + \log \frac{n_1}{\delta}})$. Here $K$ is a uniform upper bound of $s(x, y, t)\mathbb{1}\{\|x\|_2 \le R, |y| \le R\}$ for $s \in \mathcal{S}$, i.e., $\sup_{s \in \mathcal{S}} \|s(x, y, t)\mathbb{1}\{\|x\|_2 \le R, |y| \le R\}\|_2 \le K$. To this end, we have

$$\mathcal{L}(\widehat{s}) = \mathcal{L}(\widehat{s}) - \widehat{\mathcal{L}}(\widehat{s}) + \widehat{\mathcal{L}}(\widehat{s})$$

$$= \mathcal{L}(\widehat{s}) - \widehat{\mathcal{L}}(\widehat{s}) + \inf_{s \in \mathcal{S}} \widehat{\mathcal{L}}(s)$$

$$\overset{(i)}{=} \mathcal{L}(\widehat{s}) - \widehat{\mathcal{L}}(\widehat{s})$$

$$\le \mathcal{L}(\widehat{s}) - \mathcal{L}^{\mathrm{trunc}}(\widehat{s}) + \mathcal{L}^{\mathrm{trunc}}(\widehat{s}) - \widehat{\mathcal{L}}^{\mathrm{trunc}}(\widehat{s})$$

$$\le \underbrace{\sup_{s} \mathcal{L}^{\mathrm{trunc}}(s) - \widehat{\mathcal{L}}^{\mathrm{trunc}}(s)}_{(A)} + \underbrace{\sup_{s} \mathcal{L}(s) - \mathcal{L}^{\mathrm{trunc}}(s)}_{(B)},$$

where equality $(i)$ holds since $\mathcal{S}$ contains the ground truth score function. We bound term $(A)$ by a PAC-learning concentration argument. Using the same argument in [8, Theorem 2, term $(A)$], we have

$$\sup_{s \in \mathcal{S}} \ell^{\mathrm{trunc}}(x, y; s) = \mathcal{O}\left(\frac{1}{t_0(T - t_0)}(K^2 + R^2)\right).$$

Applying the standard metric entropy and symmetrization technique, we can show

$$(A) = \mathcal{O}\left(\widehat{\mathfrak{R}}(\mathcal{S}) + \left(\frac{K^2 + R^2}{t_0(T - t_0)}\right)\sqrt{\frac{\log\frac{2}{\delta}}{2n_1}}\right),$$

where $\widehat{\mathfrak{R}}$ is the empirical Rademacher complexity of $\mathcal{S}$. Unfamiliar readers can refer to Theorem 3.3 in "Foundations of Machine Learning", second edition for details. The remaining step is to bound the Rademacher complexity by Dudley's entropy integral. Indeed, we have

$$\widehat{\mathfrak{R}}(\mathcal{S}) \le \inf_{\epsilon}\frac{4\epsilon}{\sqrt{n_1}} + \frac{12}{n_1}\int_{\epsilon}^{K^2\sqrt{n_1}}\sqrt{\mathcal{N}(\mathcal{S}, \epsilon, \|\cdot\|_2)}\mathrm{d}\epsilon.$$

We emphasize that the log covering number considers $x, y$ in the truncated region. Taking $\epsilon = \frac{1}{n_1}$ gives rise to

$$(A) = \mathcal{O}\left(\left(\frac{K^2 + R^2}{t_0(T - t_0)}\right)\sqrt{\frac{\mathcal{N}(\mathcal{S}, 1/n_1)\log\frac{1}{\delta}}{n_1}}\right).$$

Here $K$ is instance dependent and majorly depends on $d$. In the Gaussian design case, we can verify that $K$ is $\mathcal{O}(\sqrt{d})$. To this end, we deduce $(A) = \widetilde{\mathcal{O}}\left(\frac{1}{t_0}\sqrt{d^2\frac{\mathcal{N}(\mathcal{S}, 1/n_1)\log\frac{1}{\delta}}{n_1}}\right)$. In practice, $d$ is often much smaller than $D$ (see for example [37], where ImageNet has intrinsic dimension no more than 43 in contrast to image resolution of $224 \times 224 \times 3$). In this way, we can upper bound $d^2$ by $D$, yet $d^2$ is often a tighter upper bound.

For term $(B)$, we invoke the same upper bound in [8, Theorem 2, term $(B)$] to obtain

$$(B) = \mathcal{O}\left(\frac{1}{n_1 t_0(T - t_0)}\right),$$

which is negligible compared to $(A)$. Therefore, summing up $(A)$ and $(B)$, we deduce

$$\epsilon_{diff}^2 = \mathcal{O}\left(\frac{1}{t_0}\sqrt{\frac{\mathcal{N}(\mathcal{S}, 1/n_1)(d^2 \vee D)\log\frac{1}{\delta}}{n_1}}\right).$$

**Gaussian Design** We only need to find the covering number under the Gaussian design case. Using (H.2), we can construct a covering from coverings on matrices $V$ and $\Sigma^{-1}$. Suppose $V_1, V_2$ are two matrices with $\|V_1 - V_2\|_2 \le \eta_V$ for some $\eta > 0$. Meanwhile, let $\Sigma_1^{-1}, \Sigma_2^{-1}$ be two covariance matrices with $\|\Sigma_1^{-1} - \Sigma_2^{-1}\|_2 \le \eta_\Sigma$. Then we bound

$$\sup_{\|x\|_2 \le R, |y| \le R}\|s_{V_1, \Sigma_1^{-1}}(s, y, t) - s_{V_2, \Sigma_2^{-1}}(x, y, t)\|_2$$

$$\le \frac{1}{h(t)}\sup_{\|x\|_2 \le R, |y| \le R}\left[\left\|V_1\psi_{\Sigma_1^{-1}}(V_1^\top x, y, t) - V_1\psi_{\Sigma_1^{-1}}(V_2^\top x, y, t)\right\|_2\right.$$

$$+ \underbrace{\left\|V_1\psi_{\Sigma_1^{-1}}(V_2^\top x, y, t) - V_1\psi_{\Sigma_2^{-1}}(V_2^\top x, y, t)\right\|_2}_{(\spadesuit)} + \left.\left\|V_1\psi_{\Sigma_2^{-1}}(V_2^\top x, y, t) - V_2\psi_{\Sigma_2^{-1}}(V_2^\top x, y, t)\right\|_2\right]$$

$$\le \frac{1}{h(t)}\left(2R\eta_V + 2\nu^{-2}R\eta_\Sigma\right),$$

where for bounding $(\spadesuit)$, we invoke the identity $\|(I + A)^{-1} - (I + B)^{-1}\|_2 \le \|B - A\|_2$. Further taking supremum over $t \in [t_0, T]$ leads to

$$\sup_{\|x\|_2 \le R, |y| \le R}\|s_{V_1, \Sigma_1^{-1}}(s, y, t) - s_{V_2, \Sigma_2^{-1}}(x, y, t)\|_2 \le \frac{1}{t_0}\left(2R\eta_V + 2\nu^{-2}R\eta_\Sigma\right)$$

for any $t \in [t_0, T]$. Therefore, the inequality above suggests that coverings on $V$ and $\Sigma^{-1}$ form a covering on $\mathcal{S}$. The covering numbers of $V$ and $\Sigma^{-1}$ can be directly obtained by a volume ratio argument; we have

$$\mathcal{N}(V, \eta_V, \|\cdot\|_2) \le Dd \log\left(1 + \frac{2\sqrt{d}}{\eta_V}\right) \quad \text{and} \quad \mathcal{N}(\Sigma^{-1}, \eta_\Sigma, \|\cdot\|_2) \le d^2 \log\left(1 + \frac{2\sqrt{d}}{\lambda_{\min}\eta_\Sigma}\right).$$

Thus, the log covering number of $\mathcal{S}$ is

$$\mathcal{N}(\mathcal{S}, \eta, \|\cdot\|_2) = \mathcal{N}(V, t_0\eta_V/2R, \|\cdot\|_2) + \mathcal{N}(\Sigma^{-1}, t_0\nu^2\eta_\Sigma/2R, \|\cdot\|_2)$$
$$\le (Dd + d^2) \log\left(1 + \frac{dD}{t_0\lambda_{\min}\eta}\right),$$

where we have plugged $\nu^2 = 1/D$ into the last inequality. Setting $\eta = 1/n_1$ and substituting into $\epsilon_{diff}^2$ yield the desired result.

We remark that the analysis here does not try to optimize the error bounds, but aims to provide a provable guarantee for conditional score estimation using finite samples. We foresee that sharper analysis via Bernstein-type concentration may result in a better dependence on $n_1$. Nonetheless, the optimal dependence should not beat a $1/n_1$-rate. □

# I  Additional Experimental Results

## I.1  Simulation

We generate the latent sample $z$ from standard normal distribution $z \sim \mathsf{N}(0, \mathsf{I_d})$ and set $x = Az$ for a randomly generated orthonormal matrix $A \in \mathbb{R}^{D \times d}$. The dimensions are set to be $d = 16, D = 64$. The reward function is set to be $f(x) = (\theta^\star)^\top x_\| + 5\|x_\perp\|_2^2$, where $\theta^\star$ is defined by $A\beta^\star$. We generate $\beta^\star$ by uniformly sampling from the unit sphere.

When estimating $\widehat{\theta}$, we set $\lambda = 1.0$. The score matching network is based on the UNet implementation from https://github.com/lucidrains/denoising-diffusion-pytorch, where we modified the class embedding so it accepts continuous input. The predictor is trained using $8192$ samples and the score function is trained using $65536$ samples. When training the score function, we choose Adam as the optimizer with learning rate $8 \times 10^{-5}$. We train the score function for $10$ epochs, each epoch doing a full iteration over the whole training dataset with batch size $32$.

For evaluation, the statistics is computed using $2048$ samples generated from the diffusion model. The curve in the figures is computed by averaging over $5$ runs.

## I.2  Directed Text-to-Image Generation

**Samples of high rewards and low rewards from the ground-truth reward model.** In Section 5.2, the ground-truth reward model is built by replacing the final prediction layer of the ImageNet pre-trained ResNet-18 model with a randomly initialized linear layer of scalar outputs. To investigate the meaning of this randomly-generated reward model, we generate images using Stable Diffusion and filter out images with rewards $\ge 0.4$ (positive samples) and rewards $\le -0.4$ (negative samples) and pick two typical images for each; see Figure 7. We note that in real-world use cases, the ground-truth rewards are often measured and annotated by human labors according to the demands.

**Training Details.** In our implementation, as the Stable Diffusion model operates on the latent space of its VAE, we build a 3-layer ConvNet with residual connections and batch normalizations on top of the VAE latent space. We train the network using Adam optimizer with learning rate $0.001$ for $100$ epochs.

## I.3  Decision-Diffuser [1]

We replicate the results in Decision Diffuser [1] under (Med-Expert, Hopper) setting. In Decision Diffuser, the RL trajectory and the final reward are jointly modeled with a conditioned diffusion model. The policy is given by first performing reward-directed conditional diffusion to sample a

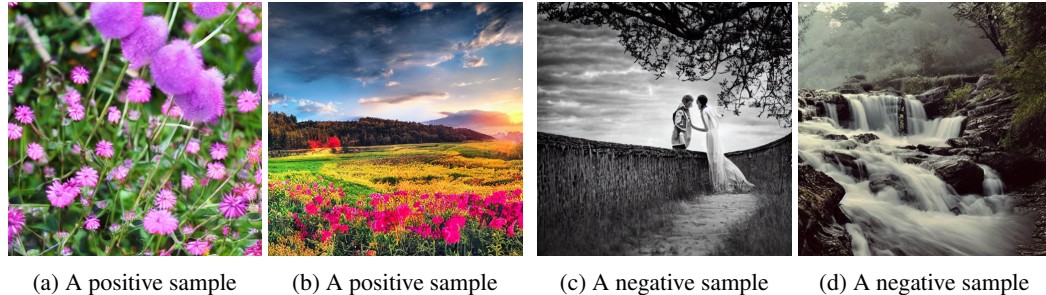

(a) A positive sample      (b) A positive sample      (c) A negative sample      (d) A negative sample

Figure 7: Random samples with high rewards and low rewards.

trajectory of high reward and then extracting action sequences from the trajectory using a trained inverse dynamics model. We plot the mean and standard deviation of the total returns (averaged across 10 independent episodes) v.s. the target rewards in Figure 8. The theoretical maximum of the reward is 400. Therefore, trajectories with rewards greater than 400 are never seen during training. We observe that when we increase the target reward beyond 320, the actual total reward decreases. According to our theory, as we increase the reward guidance signal, the condition effect becomes stronger but the distribution-shift effect also becomes stronger.

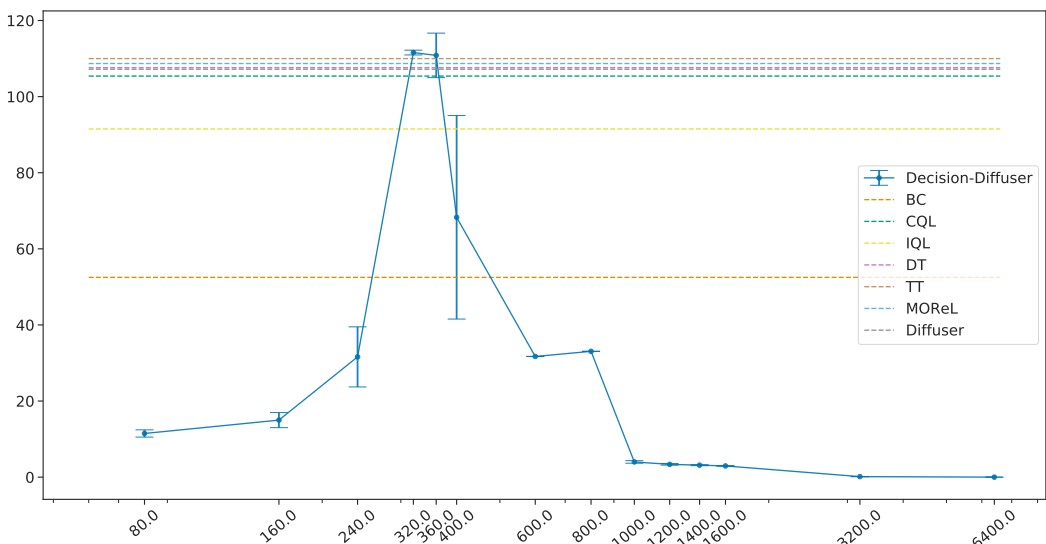

Figure 8: Total Reward v.s. Target Reward for Decision Diffuser (Med-Expert, Hopper) setting.

