# OpenReview forum: "Reward-Directed Conditional Diffusion: Provable Distribution Estimation and Reward Improvement"
_NeurIPS.cc/2023/Conference — NeurIPS 2023 poster_

### Official Review · Reviewer_uwg2 · 2023-07-04

**Soundness:** 2 fair
**Presentation:** 3 good
**Contribution:** 2 fair
**Rating:** 6
**Confidence:** 2

**Summary:**

The paper addresses conditional generation with reward-conditioned diffusion models. They propose to learn a reward function from a small subset of labeled data. The paper aims to answer an intriguing research question: "How can we reliably estimate the reward-conditioned distribution through diffusions and balance the trade-off between the reward signal and generating quality?" Additionally, the paper claims that the reward-conditioned diffusion model implicitly learns the latent subspace representation of x.

**Strengths:**

Strength:
1)  The paper is well written and the problem they try to tackle is interesting.
2) The paper provides an insightful theory for conditional distribution learned through a reward-based diffusion model


**Weaknesses:**

1) The paper lacks a comparison to other similar models, such as classifier-guided diffusion models.

2) The assumptions made for the theoretical work appear to be overly simplified.

**Questions:**

Some suggestions:
1) It would be good to add what each color represents in Figure 2.
2) I think the main paper should include at least a summary of related work so that reader can have a better picture of where this work stands when compared to the existing works.
3) In Figure 1, in the block for the diffusion model, X_0 is represented as the noise, I think it is better to keep the same notation as in other literature to avoid confusion.

Questions
1) Regarding Figure 6, it is mentioned that those are picked examples, does it mean the presented results are "cherry-picked" and not generalizable? also, I think here y represents the colorfulness level ( as the ground truth reward model favors colorful images), there seems the context is also changed, not only the color, do you have any insight on that?
2) The model seems very similar to the classifier-guided diffusion model, but it never compared to that in the experiments, any reason for that?
3) I am wondering if it is possible to train the reward function together with the diffusion model instead of pertaining to the reward network.
4)  The paper assumes generated data is a linear transformation of the latent z (assumption 3.1), I am wondering if this assumption is too strong and does not hold in general.
5) Assumption 3.2 assume a linear reward, however, the ground truth reward in the experiment used the imageNet with a randomly initiated final prediction layer.  I am not sure how to interpret this part.
6)  After reading the paper, I am still confused about the assumption that the model implicitly learns the latent subspace representation of x.

Some suggestions:
1) It would be beneficial to add a legend in Figure 2 that explains the meaning of each color.
It is recommended to include a summary of related work in the main paper to provide readers with a better understanding of where this work stands in comparison to existing literature.
2) In Figure 1, the block representing the diffusion model uses X_0 to denote noise. It may be clearer to use consistent notation with other literature to avoid confusion.


Questions:

1) Regarding Figure 6, it is mentioned that the examples shown are picked intentionally. Does this imply that the presented results are cherry-picked and may not be generalizable? Additionally, it appears that y represents the colorfulness level (as the ground truth reward model favors colorful images), but there seems to be a change in context as well. Do you have any insights on this?
2) The model appears to be quite similar to the classifier-guided diffusion model, yet there is no comparison with it in the experiments. Is there a specific reason for this omission?
3) I'm curious if it is possible to train the reward function concurrently with the diffusion model instead of relying on a separate reward network.
4) The paper assumes that the generated data is a linear transformation of the latent z (assumption 3.1). However, I'm wondering if this assumption is overly strong and may not hold in general.
5) Assumption 3.2 assumes a linear reward, but the ground truth reward used in the experiments involved ImageNet with a randomly initiated final prediction layer. I'm unsure how to interpret this aspect.
6) Even after reading the paper, I'm still unclear about the assumption that the model implicitly learns the latent subspace representation of x.



**Limitations:**

The paper did not discuss any limitations.
The paper has no potential negative societal impact.

---

> ### Author Rebuttal · Authors · 2023-08-10
>
> >**Q1**. The paper lacks a comparison to other similar models, such as classifier-guided diffusion models.
>
> **A1**. Alg 1 is not an alternative to classifier-guided diffusion. Instead, it is a simplification of it and also generalized to continuous reward and semi-supervised learning. Please refer to "2.Comparison" in [our rebuttal](https://openreview.net/forum?id=58HwnnEdtF&noteId=npWKushsuE) for a detailed comparison between Alg.1 and guidance methods. Rather than proposing an algorithm that outperforms the existing ones like classifier(-free)-guided diffusion, the purpose of Alg.1 is to give a formal mathematical statement of the conditional diffusion procedure with rigorous mathematical guarantees.
>
> >**Q2**. The assumptions appear to be overly simplified.
>
> **A2**. There is a misunderstanding. Our assumptions are in fact mild and general. We allow practical encoder-decoder architecture for score matching, data with latent representation, nonlinear reward, noparametric reward learning, nonparametric distribution with mild regularity conditions.
>
> Next we discuss two of our assumptions: (1) low-dimensional linear representation that x = Az. (2) parametric/nonparametric form of the ground-truth reward and regularities of data distribution.
>
> For (1) , our theorems hold in either case where data lies in an unknown subspace spanned by columns of $A$ with a smaller column dimension, or the case $A = I_D$ meaning that data is full-dimensional. Thus, our theory adapts to data distributions with an arbitrary intrinsic dimension. Our analysis also applies to nonlinear data through kernel transformation. For example,  when data $x = \phi^{-1}(z)$ for a known mapping $\phi$ (for example if we have kernel or feature map) and latent variable $z$, we consider linear conditional diffusion on $z = \phi(x)$ and our results immediately apply with a simple transform.
>
> For (2), in section 3 we present in detail the theorems for a parametric setting of reward and data distribution; and then in Section 3.3 and Appendix G an extension to nonparametric setting that allows nonlinear reward and general data distributions.
> In the parametric setting, reward is assumed to be linear and x is assumed to be Gaussian. As the first theory for conditional diffusion, this parametric configuration is a go-to setting to study since it builds up the theoretical foundation and gives most insights for other advanced settings such as logistic, kernel, neural network (NTK) models. The nonparametric setting further gives generality of our theorems so that it covers the practical scenario where deep ReLU networks are adopted to approximate the reward and score.
>
> >**Q3**. It would be beneficial to add a legend in Figure 2 that explains the meaning of each color and include a summary of related work in the main paper
>
> **A3**. Figure 2 is updated with legend ([preview is here](https://openreview.net/attachment?id=npWKushsuE&name=pdf)). Thanks for your suggestion. Due to the space limit, we had to defer related work to Appendix A. We will try to move it back in the final paper.
>
> >**Q4**. In Figure 1, the block representing the diffusion model uses X_0 to denote noise. It may be clearer to use consistent notation with other literature to avoid confusion.
>
> **A4**. The block in Figure 1 represents the backward process of diffusion models (note the left arrow in the superscript), which is not to be confused with the often-seen forward process. Therefore, the starting point of the backward process is pure noise. This notation is also adopted by some other papers in diffusion literature such as [1].
>
> [1]  Chen et al. Score approximation, estimation and distribution recovery of diffusion models on low-dimensional data.  arXiv:2302.07194, 2023.
>
> >**Q5**. Figure 6, are examples cherry picked? why does context changes with colorfulness?
>
> **A5**. The examples are typical and we didn’t cherry pick. The reward was randomly constructed for generality. We will release the model checkpoints and our code upon acceptance.  During rebuttal, we also conducted a [new experiment](https://openreview.net/attachment?id=npWKushsuE&name=pdf) on RL task.
>
> >**Q6**. Is it possible to train the reward function with the diffusion model instead of relying on a separate reward network.
>
> **A6**. In practice it’s possible. In theory, we use a separate reward trained on labeled data that is independent from unlabeled data, for ease of theoretical analysis and better separating the error from reward training and diffusion training in our guarantees.
>
> >**Q7**. Assumption 3.2 assumes a linear reward, but the ground truth reward used in the experiments involved ImageNet with a randomly initiated final prediction layer. I'm unsure how to interpret this aspect.
>
> **A7**. Our main paper only presented results with linear reward in detail due to page limit, but our theory extends to nonparametric nonlinear rewards and general distributions (section 3.3). Our synthetic experiment in section 4.1 directly tested the linear reward, and our imagenet experiment tested with nonlinear reward which can be interpreted using the neural tangent kernel theory. During rebuttal, we also provided a new RL experiment where y is the value of policy and depends on x via complicated nonlinear relations.
>
> >**Q8**. I'm still unclear about the assumption that the model implicitly learns the latent subspace representation of x.
>
> **A8**. There is a misunderstanding. We never assume “The model implicitly learns the latent subspace representation of x”. Instead, we proved it as one of our main results, i.e.,  (3.2) in Theorem 3.5 where the subspace angle between the ground truth $A$ and the learned one $V$ is upper bounded by $\frac{1}{\sqrt{n_1}}$, proving that diffusion model learns the latent subspace representation of $x$. The implication is that the DM is able to learn the true natural space of data and thus generate high-fidelity samples.

---

> > ### Author Response · Authors · 2023-08-15
> > **Author's Follow-up**
> >
> > Dear Reviewer,
> >
> > We want to follow up to gently check in your opinions on our rebuttal posted one week ago. A brief summary of our rebuttal to your concerns is:
> > - We compared our method to guidance methods in detail in 2. Comparison of [our rebuttal](https://openreview.net/forum?id=58HwnnEdtF&noteId=npWKushsuE). It's evident from the comparison that Alg.1 and its analysis captures the essence of both classifier guidance and classifier free guidance methods.
> > - "The assumptions appear to be overly simplified" is a misunderstanding. In **Q2&A2**, we gave a fine-grind breakdown of what assumptions we made and what results we include in our paper.
> >
> > By the way, we added a [new experiment](https://openreview.net/attachment?id=npWKushsuE&name=pdf) on solving RL tasks via Alg.1 to showcase its versatility. Please let us know if you have further suggestions. Thank you very much!

---

### Official Review · Reviewer_mN9F · 2023-07-06

**Soundness:** 3 good
**Presentation:** 3 good
**Contribution:** 2 fair
**Rating:** 6
**Confidence:** 3

**Summary:**

In this work, authors explore the problem of reward-directed generation using conditional diffusion models in a semi-supervised learning setup. More specifically, they consider a dataset which has a small subset of it labeled with rewards and the majority of it unlabeled. Using the small labeled subset, they first train a reward approximator with regression, then use the learned reward approximator to label the unlabeled portion of the dataset with pseudo labels. Finally, they train the conditional diffusion model using the samples and their pseudo labels. Authors present theoretical explanations for conditioned diffusion models and reward improvement guarantees for reward-conditioned generations. Finally, experimental results are provided that demonstrated the findings of the theory.


**Strengths:**

Overall Strengths:
1. This paper is very well-organized and well-written and I was able to follow along fairly easily. So, I highly commend the authors for the great job they have done. In particular, all the sections leading up to the main theory are organized well and give a nice background and trajectory to where the theory lies.
2. The high-level idea of theoretically studying how reward-conditioned distribution of diffusion models change and how to balance reward to trade-off sample quality and reward maximization is well-worth pursuing.
3. Theory is well-driven and thoroughly discussed.


**Weaknesses:**

Overall Weaknesses:
1. One of the main questions that this paper aims to answer is “How to balance the reward signal and distribution-shift effect to have high-reward and high-quality samples?” which is what I was most excited to learn about. However, I'm not confident that I truly got an answer to this question.
2. The experimental setup is limited and the computed metrics do not thoroughly cover the theoretical findings of the papers.
3. Although figure 2 looks very nice, I’m not sure if it has any added value in the main body of the paper.


**Questions:**

1. How will the results generalize to settings where data doesn’t have a latent linear representation?
2. How will the results of the current model (conditional diffusion model) compare to using an unconditional diffusion model with classifier guidance to achieve high rewards?
3. For text-to-image generation, it is said that higher reward refers to more vividly colored images. Do you have a numerical metric for this?


**Limitations:**

In addition to the points provided in the weaknesses section, I believe a deeper effort on the experiments section would be of tremendous value for this work. For me, the experimental results fall short in supporting the theory of the paper and I’m not sure if I’m convinced the full message is conveyed.

---

> ### Author Rebuttal · Authors · 2023-08-10
>
> Thank you for your valuable comments!
>
> >**Q1**. “How to balance the reward signal and distribution-shift effect to have high-reward and high-quality samples?” Experimental setup is limited
>
> **A1**. Our paper provides the first theory for conditional diffusion and use of reward-conditioned diffusion to generate better samples. There is a rich body of empirical works using CDM for reward maximization in various contexts [1].
>
> In general, designing an explicit formula for the optimal target is difficult, as the interplay between the guidance level and the distribution shift is data dependent and complicated. There is unlikely a one-fits-all solution, which is why we need theory research. Theorem 3.6 is the first rigorous result characterizing this complicated tradeoff under very general assumptions. In practice, one can use a simple doubling trick (ie binary search) to tune the target value since it is only 1-dim, at a cost of only a log factor in time.
>
> Our experiments are for the purpose of illustration and complements the theory. For rebuttal, we add a [new experiment](https://openreview.net/attachment?id=npWKushsuE&name=pdf) on RL and showcased the use of CDM with varying target values to maximize value. We welcome any further suggestions on the experiment part. But we want to keep our focus on deep learning theory, more extensive experiments would better fit an application paper.
>
> [1] https://arxiv.org/abs/2211.15657
>
>
> >**Q2**. Although figure 2 looks very nice, I’m not sure if it has any added value in the main body of the paper.
>
> **A2**. Thank you for the comments. Figure 2 is closely related to our method and results, but we realize that it lacks annotations and might have confused you. We have updated Figure 2 with clear annotations and explanations ([a preview is here](https://openreview.net/attachment?id=npWKushsuE&name=pdf)).
>
> Subfigure (a) and (b) illustrate the distribution shift when extrapolating reward prediction and the distribution shift in extrapolating diffusion respectively, these two are the key components in the error decomposition shown in Theorem 3.6. Subfigure (a) corresponds to term $\mathcal{E}_1$ and Subfigure (b) depicts the on-support situation corresponding to $\mathcal{E}_2$, $\mathcal{E}_3$ occurs in the space perpendicular to the subspace in (b) and is not explicitly drawn out . We updated the text annotations wrapped around figure 2 pointing its connection to theory. (c) illustrates the architecture of score matching network we analyzed.
>
> >**Q3**. How will the results generalize to settings where data doesn’t have a latent linear representation?
>
> **A3**. Our theory applies to general data distributions. We show that the conditional diffusion model naturally adapts to data distribution (high-dimensional data simply corresponds to the case where $A = I_D$, and when data lies in an unknown subspace, it corresponds to $A$ with a smaller column dimension).
>
> Our results also apply to nonlinear data such as Fourier series or data from kernel spaces with a known basis transformation. For example, when data $x = \phi^{-1}(z)$ for a known mapping $\phi$ and latent variable $z$, we consider linear conditional diffusion on $z = \phi(x)$ and our analysis immediately applies with an additional $\phi^{-1}$ transform.
>
> >**Q4**. How will the results of the current model (conditional diffusion model) compare to using an unconditional diffusion model with classifier guidance to achieve high rewards?
>
> **A4**. Our model can be viewed as a special case of classifier-free guidance. It is simpler than classifier guidance and doesn’t need multiple classifier $c_t$’s. Please see "Q2.Comparison" in [our renuttal](https://openreview.net/forum?id=58HwnnEdtF&noteId=npWKushsuE) for detailed discussion.
>
> >**Q5**. For text-to-image generation, it is said that higher reward refers to more vividly colored images. Do you have a numerical metric for this?
>
> **A5**. In our text-to-image generation experiment, a higher reward only "favors" more colorful images, that is, we observe a high correlation of large reward and colorfulness. As our target is to maximize the reward (which is not exactly the same as colorfulness), we only need to track the reward values of the generated samples. Therefore, the numerical metrics can be evaluated by running samples through our ground-truth reward model, which only *correlates* with more vividly colored images.

---

> > ### Comment · Reviewer_mN9F · 2023-08-14
> >
> > I appreciate the authors' thorough response.
> >
> > After a careful re-evaluation of the work, considering the input from other reviewers and taking into account the authors' rebuttal efforsts, I have decided to increase my rating. I particularly find the newly added experiment on all baselines helpful.

---

> > > ### Author Response · Authors · 2023-08-15
> > >
> > > Thank you for your swift and positive feedback on our rebuttal. We appreciate your effort in reviewing and discussion period, which really helps us improve.

---

### Official Review · Reviewer_ZNG8 · 2023-07-06

**Soundness:** 3 good
**Presentation:** 2 fair
**Contribution:** 3 good
**Rating:** 6
**Confidence:** 2

**Summary:**

This paper presents an approach to generation using diffusion models augmented with a reward function. It does so by setting up a semi-supervised learning setup, where the reward function is learned from a small set of data. The reward is then used to learn a reward conditioned score function, which is subsequently used to generate data conditioned on requested reward.
Assuming a linear subspace, the analysis approaches this setting through the lens of linear bandits, and characterizes the error or suboptimality in terms of regret to the target or requested reward, off-distribution error, and on-distribution error.

Experiments evaluate the above interplay, and how requesting higher rewards leads to distribution shift in the generated samples. Further experiments show how pre-trained models can be adapted to generate reward-conditioned samples.

**Strengths:**

* Presents a practical way to add subjective rewards to generate data beyond concrete prompts like text
* Analyzes this reward conditioned generation setting and identifies the interplay between rewards and distribution shift.
* Experimentally verifies the claims made in the analysis.

**Weaknesses:**

* The paper does not discuss a practical manner in to identify when the generative model starts deviating from the training distribution.
* Comparing the diffusion model's error to a linear bandit setting is interesting, but the exact connection between the two seems a bit murky. The training process does not seem to take advantage of any bandit learning algorithms.
* While it shows that the model can be adjusted to arbitrary rewards, it does not showcase a practical use case.
* The technical novelty is unclear.

=============================
### Post-Rebuttal
* The author's rebuttal and other reviews have made the technical contribution of the paper abundantly clear.
* Additionally, the connection to _offline_ bandits is more evident and does add value.
* The additional RL experiment grounds the claims made in the paper more concretely than the previous experiments.

As such, the majority of my concerns have been allayed with the author response.

**Questions:**

* Section 2.1 gives the scale of $\sigma$ but does not mention any such constraint about $f^*$. Section 4.2 constructs a random reward, but this setting also makes no mention of the scale of the reward. How does the scale of the reward affect learning?
* If someone were to try and reproduce these results, how would they go about it? Perhaps an experiment that specifically tries to maximize some specific property that is hard to communicate through a text prompt could be shown here, to communicate the effectiveness of this approach, as well as to assist reproducibility.
* The labeled data is based on the CIFAR-10 dataset. Is the unlabeled dataset from a corresponding distribution? Does it have similar resolution, size, and kinds of pictures?
* The paper states `To optimally choose a target value, we must trade off between the two counteractive effects.` Are there any practical methods to do so?
* Reinforcement learning faces the problem of reward design. The analysis done in this paper could be useful for this problem of reward design. It would be useful to deepen the discussion on reward design and reference some related work  [1, 2]

### References
[1] Booth, S., Knox, W.B., Shah, J., Niekum, S., Stone, P. and Allievi, A., 2023, June. The perils of trial-and-error reward design: misdesign through overfitting and invalid task specifications. In Proceedings of the AAAI Conference on Artificial Intelligence (Vol. 37, No. 5, pp. 5920-5929).

[2] Knox, W.B., Allievi, A., Banzhaf, H., Schmitt, F. and Stone, P., 2023. Reward (mis) design for autonomous driving. Artificial Intelligence, 316, p.103829.

**Limitations:**

while the proposed approach opens up avenues to communicate and generate data using feedback other than text prompts, it does not sufficiently address problems that might arise from such freeform feedback.

---

> ### Author Rebuttal · Authors · 2023-08-10
>
> >**Q1**. The paper does not discuss a practical manner in identifying when the generative model starts deviating from the training distribution.
>
> **A1**. Our focus is theory and implications are listed in “impact and novelty” in [our rebuttal](https://openreview.net/forum?id=58HwnnEdtF&noteId=npWKushsuE). We agree that to identify a proper target so that generated samples are deviating from training distribution towards higher reward but not deviating too much is the key of success for empirical applications. In practice, the guidance level is viewed as a tuning parameter, and optimal choice is achieved by hyperparameter tuning. In theory, designing an explicit way for identifying the optimal target is difficult, as the interplay between the guidance level and the distribution shift is data dependent and complicated (see our discussion following Theorem 3.6 (Line 214)). There is unlikely to be a general rule of thumb or a one-fits-all way to identify.
>
> >**Q2**. Comparing the diffusion model's error to a linear bandit setting is interesting, but the exact connection between the two seems a bit murky. The training process does not seem to take advantage of bandit algorithms.
>
> **A2**. Thank you for bringing up this confusion. We agree that the theoretical connection between CDM and bandit is intriguing and we are the first to observe this potential connection.  A detailed discussion:
>
> Our problem is similar to *offline* bandits in two ways: (1) both problems deal with offline data of the form {(x,y)}; (2) the goal is to find new x with improved reward value y. Given its offline nature, it does not benefit from any online bandit exploration techniques such as UCB or Thompson sampling.
>
> Note that our theory is not about linear bandits, and we never established any form of equivalence. The statistical error of condition diffusion consists of several parts (Theorem 3.6): $\mathcal{E}_1$ (due to reward function estimation) and $\mathcal{E}_2, \mathcal{E}_3$ (due to conditional score matching)
> Among these terms, only a single term $\mathcal{E}_1$ resembles the suboptimality gap in off-policy bandit/RL (if we pick target value to be the max value). Our full analysis of reward-directed DM is much more complex beyond $\mathcal{E}_1$. In addition, our analysis is not limited to linear reward, but allows general nonparametric reward (Section 3.3).
>
> Further, in our [new experiment](https://openreview.net/attachment?id=npWKushsuE&name=pdf), we tested CDM on an RL problem and obtained improved reward performance close to the best known off-policy RL benchmark. This observation is also consistent with our theory.
>
> >**Q3**. While it shows that the model can be adjusted to arbitrary rewards, it does not showcase a practical use case.
>
> **A3**: Our primary focus is to establish thoery for CDM, especially for reward maximization. Many papers have already showcased the practical usage of CDM for generating high-reward samples in various contexts, not limited to image generation [3] but also control and RL [1,2]. We were motivated by those empirical successes and decided to go deep in theory. To this end, we formulate the practical scenario of semi-supervised learning and provided Alg. 1 as a meta algorithm (see "2. Comparison" in [our rebuttal](https://openreview.net/forum?id=58HwnnEdtF&noteId=npWKushsuE) for its close relation to guidance methods).  Our results are not limited to a specific use case, instead they provide insights on diverse applications.
> - [1] https://arxiv.org/abs/2211.15657 (2022)
> - [2] https://arxiv.org/abs/2302.01877 (2023).
> - [3] https://arxiv.org/abs/2305.13301 (2023).
>
> >**Q4**. The technical novelty is unclear.
>
> **A4**. The theory of (conditional) diffusion models is widely open. There are very limited results providing statistical theories. To the best of our knowledge, we are the first to give theory for conditional diffusion models and connect them to bandits. In our theory, we provide novel analysis on conditional score estimation from both a parametric and nonparametric point of view. Moreover, in Theorem 3.6, we developed an oracle-type decomposition of the suboptimality gap into three terms, indicating the trade-off on the guidance level $a$ and distribution shift. These results are all derived by using novel analysis and new techniques.
>
> >**Q5**. Section 2.1 gives the scale of sigma but does not mention any such constraint about $f^*$ .
>
> **A5**. In section 2.1, constraint on the scale of $f^*(x)$ is imposed by the assumption on $g*$ (on-support component of $f^*$) and the distribution of $z$ (latent of $x$). In the non-parametric section, $f^*(x)$ has a similar constraint. The scale of reward reflects on term $\mathcal{E}_1$ in Theorem 3.6, which grows linearly with the scare of $f^*$. In practice, It is common to normalize the reward by preprocessing data before training so that the scale does not influence learning.
> In both of our experiments, the scales of the reward function fall into normal ranges. Please refer to Figure 2 (Left, Right) for the training distributions of the rewards. In the new RL experiment (see general response above), the rewards are normalized to line in [0,1].
>
> >**Q6**. Reproducibility of experiments.
>
> **A6**. Our main results are Theorem 3.5, 3.6 and their extensions to nonparametric setting in Section 3.3, with proofs fully fleshed out in appendix and easily reproducible. While the experiment is only for illustration, we will release code and demos in the final paper. Any user can supply a reward model oracle and directly use our code to generate high-reward data.
>
>
> >**Q7**. The labeled data is based on the CIFAR-10 dataset. Is the unlabeled dataset from a corresponding distribution? Does it have similar resolution, size, and kinds of pictures?
>
> **A7**.The unlabeled dataset is LAION-5B. LAION-5B dataset has various image resolutions and sizes and covers various scenes. Both datasets correspond to natural images.

---

> > ### Comment · Reviewer_ZNG8 · 2023-08-14
> > **Thank you for the Response**
> >
> > I thank the authors for the detailed response.
> >
> > The confusion about the connection to linear bandits likely arose from this sentence in the paper on line 67:
> > ```
> > In the case of a linear reward model, we show that the regret mimics the off-policy regret of linear bandits with full knowledge of the subspace feature.
> > ```
> >
> > I agree that in the offline setting, typical bandit algorithm approaches are not applicable.
> >
> > Additionally, my main concern about technical novelty has been allayed. The experiment in the RL domain has especially helped to ground the idea in a concrete problem.
> > I have raised my score accordingly.

---

> > > ### Author Response · Authors · 2023-08-15
> > >
> > > Thank you for your prompt feedback on our rebuttal. The sentence you pointed out does need more clarity to avoid confusion. We will modify it based on the facts in **A2** to
> > > ```
> > > Given a target reward value, we analyze the statistical error of reward-directed generation, measured by the difference between the target value and the average reward of the generated population. In the case of a linear reward model, we show that this error includes the suboptimality gap of linear off-policy bandits with full knowledge of the subspace feature, if taking the target to be the maximum possible.
> > > ```
> > > In addition, thanks for your especially positive comment on our new experiment. Your positive rating flipped from the negative is very encouraging and reassuring to us :)

---

### Official Review · Reviewer_umuF · 2023-07-09

**Soundness:** 3 good
**Presentation:** 2 fair
**Contribution:** 3 good
**Rating:** 6
**Confidence:** 3

**Summary:**

The paper addresses the problem of conditional generation with diffusion models in a self-supervised setting, where the conditional generation is guided by a learned regressor on the small labeled subset. This is referred to as reward-directed conditional diffusion. Assuming the inputs have a latent linear representation, it is shown that reward-conditioned diffusion models implicitly learn this latent representation. Further, assuming a linear reward model is used, it is shown that reward-conditional generation can be viewed as off-policy bandit learning in latent feature space. The theory is also extended to non-nonparametric reward and score functions. Experiments on synthetic and text-to-image data support the theory.

**Strengths:**

Steering generate models (especially diffusion models) towards generating samples with desirable properties is a topic of wide interest. The semi-supervised setting is a particularly important special case of that, which comes up in many real-world applications. As the authors note, there is not much theoretical work in this space yet, making this a valuable contribution.

I found the theoretical analysis to be quite insightful. I particularly like the analysis regarding the trade-off between distribution shift and reward maximization. The simulation experiment seems to align well with the theory. I also appreciate that the theoretical analysis extends to more general reward and score functions.


**Weaknesses:**

I found some parts of the paper a bit difficult to read. First, the motivation and derivation of the score network architecture is unclear without reading the reference ([8]). For example, it is not obvious that this functional form follows from the linearity assumption of x = Az. Second, a lot of notation is not introduced, e.g. $k$ in Eq. 2.3, $x_\parallel$ and $x_\perp$ in Assumption 3.2. This makes some equations difficult to understand. Third, it does not become clear until halfway through the paper why the pseudo-labeller is called a reward function. Perhaps a forward reference would help to clarify this.

I am also not sure what the practical implications are. I could have (qualitatively) predicted the results for text-to-image generation without the theory, simply because you are asking the model to extrapolate beyond the training data. The quantitative results also only make qualitative statements.

Apart from that, I found the experimental setup of using a random reward model a bit strange.

Minor comments:
- Given the similarity of Fig. 2c with Fig 2 in [8], citing [8] is probably warranted ("Figure adapted from [8]")

**Questions:**

1. It is not clear to me why Algorithm 1 (especially Eqs 2.2 and 2.3) is necessary. Why couldn't you use either classifier-free or classifier-based guidance? Why does Algorithm 1 offer "theoretical cleanness"?
2. I do not understand why Assumption 3.2 is required. To me, it feels more like a user choice to penalize off-support data in the reward as this should also be reflected in the unconditional score (recall that $\nabla_x \log p(x|y) = \nabla_x \log p(y|x) + \nabla_x \log p(x)$). Could you clarify this?
3. In the experiments, what is the reward distribution of the training data? At which reward values are you asking the model to extrapolate?

Minor comments:
- In the problem setup section, why is $1 > \sigma$ necessary?

**Limitations:**

The authors have clearly laid out the assumptions for the theoretical analysis. I do not see any major limitations within this scope.

---

> ### Author Rebuttal · Authors · 2023-08-10
>
> Thanks for your thoughtful and insightful review.  We’ve revised our paper to clarify the notations and added more explanations around the score network according to your suggestions.
>
> >**Q1**. What are the practical implications. I could have (qualitatively) predicted the results for text-to-image generation without the theory.
>
> **A1**. Our theory is not limited to understanding image generation, but also covers applying diffusion for RL and control tasks (see [1,2] and our [new experiment](https://openreview.net/attachment?id=npWKushsuE&name=pdf). You are absolutely right that there’s an extrapolation from training data happening, but there are more questions that we answered by our theorems.
>
> Implications of theory:
> - Fidelity of generated samples: conditional score matching network with architecture (2.8) accurately recovers any latent subspace structure in data.
> - Trade-off between high target reward $a$ and large distribution-shift and a comprehensive exposition of the distribution-shift associated with $a$.
> - Low-dimensionality dependency: error coming from reward learning only depends on the subspace dimension $d$, spot a relation to offline bandit/RL.
> - Nonparametric results beyond linear settings
>
> Please refer to the "impact and novelty" section in [our rebuttal](https://openreview.net/forum?id=58HwnnEdtF&noteId=npWKushsuE)  for a more detailed exposition on this question.
>
> >**Q2**. the experimental setup of using a random reward model a bit strange.
>
> **A2**: Our paper focuses on general reward-guided diffusion rather than the specific application of images, thus we use a random function for reward for generality. In particular, in the image example, we constructed the reward by using a random linear projection layer on top of an ImageNet pre-trained ResNet model. It can reflect real-world applications because (1) in practice rewards may be unknown and require expensive data collection processes; (2) the pre-trained ResNet models are believed to extract useful representations for images that are semantically meaningful, thus it’s very likely a real-world reward is an easy (MLP) function of  these representations. In addition, we provided a [new experiment](https://openreview.net/attachment?id=npWKushsuE&name=pdf) on RL where the reward is predefined in the environment but has to be learned through jointly modeling the trajectories and the final rewards using a conditional diffusion model.
>
> >**Q3**. Why Algorithm 1 (especially Eqs 2.2 and 2.3) is necessary. Why couldn't you use either classifier-free or classifier-based guidance? Why does Algorithm 1 offer "theoretical cleanness"?
>
> **A3**: Alg. 1 does offer cleanness to our analysis. To clarify, classifier and classifier-free guidance can be formulated by the exact same backward sampling as (2.3), by plugging in a similar but different score from what is learned by (2.2). Comparison [link] discusses the relation between Alg.1 and guidance methods. Alg.1 captures the essence of both classifier guidance and classifier free guidance, i.e. to estimate the conditional score $\nabla \log p_t (x_t | y)$, but with an ease of putting aside other less-essential components, such as the multiple classifiers in classifier guidance and the mix of conditioned and unconditioned scores in classifier-free guidance.
>
> >**Q4**. Why Assumption 3.2 is required. To me, it feels more like a user choice to penalize off-support data in the reward as this should also be reflected in the unconditional score.
>
> **A4**. Assumption 3.2 configures the ground truth reward by its $g$ and $h$ components. Linearity in $g$ is required for bounding the term $\mathcal{E}_1$ in Theorem 3.6. As for the $h$ component, we make it explicit otherwise the reward could be ill-defined. We agree that $h$ is for penalizing off-support data in the reward,  and it can be problem dependent. Our result (Theorem 3.6) still holds with any choice of $h$, depending on problem nature or user choice.
>
> Yes, the off-support portion of x will be reflected in its score $\nabla_x \log p(x|y)$. If the conditional diffusion model learns the score $\nabla_x \log p(x|y)$ well, then generated data form it should have little off-support component, which is what we proved in Theorem 3.5 by upper bounding $x^{\perp}$ showing $x$ has high fidelity to the support. The ground-truth reward in Assumption 3.2 models a combination of data utility and fidelity.
>
> >**Q5**. In the experiments, what is the reward distribution of the training data? At which reward values are you asking the model to extrapolate?
>
> **A5**: In our synthetic experiment (Section 4.1), the unconditioned reward distribution follows the standard Gaussian distribution $N(0, 1)$. When we set target reward value $\ge 3$, this falls outside the $3\sigma$ region of the Gaussian distribution, and we view any reward level beyond 3 as extrapolation. The visualization of the reward distribution can be found in Figure 2 (Left) in our supplementary pdf file.
>
> In our image generation experiment (Section 4.2), the mean and the standard deviation of the reward of the training data are mean -0.9113 std = 0.4130
> When we set target reward value $0.4$ (outside $3\sigma$), we can view it as asking the model to extrapolate. The visualization of the reward distribution can be found in Figure 2 (Right) in our supplementary pdf file.
>
> In our new RL experiment, the mean and the standard deviation of the reward are
>  mean =294.8 std = 91.2. (Note: This is an offline dataset generated by human experts, so it is higher than all reported offline RL algorithms.)  The visualization of the reward distribution can be found in Figure 1 (Right) in our supplementary pdf file.
>
> >**Q6**. In the problem setup section, why is 1>sigma necessary?
>
> **A6**: We let $\sigma<1$ for simplicity. It is only a scaling constant. If $\sigma>1$, Theorem 3.6 would still hold and the term $\mathcal{E}_1$ will linearly scale up with $\sigma$.

---

> > ### Author Response · Authors · 2023-08-10
> > **Locating Figures Mentioned in Q5 & A5**
> >
> > We want to remind you that **Figure 1** and **Figure 2** mentioned in our reponse **A5** are located in our [new experiment page](https://openreview.net/attachment?id=npWKushsuE&name=pdf), in case there's any confusion on where to find them.

---

> > > ### Author Response · Authors · 2023-08-15
> > > **Authors' Follow-up**
> > >
> > > Dear Reviewer,
> > >
> > > We want to follow up to gently check in your opinions on our rebuttal posted one week ago. To give a quick recap, we explained in detail in [our rebuttal](https://openreview.net/forum?id=58HwnnEdtF&noteId=npWKushsuE) on 1. implications of theory 2. comparison to guidance methods. In addition, we added a [new experiment](https://openreview.net/attachment?id=npWKushsuE&name=pdf) on solving RL task via Alg.1 to showcase its versatility.
> > >
> > > Please let us know if we have addressed your concern and if you have further suggestions. Thank you very much!

---

> > > > ### Comment · Reviewer_umuF · 2023-08-15
> > > >
> > > > Thank you very much for the comprehensive reply. I really appreciate the additional explanations and experiments.
> > > >
> > > > The rebuttal confirms my initially positive take on the paper. Thus, I will maintain my score. In addition, I am slightly raising my confidence score given the additional clarity.

---

### Author Rebuttal · Authors · 2023-08-10

We would like to thank all the reviewers for valuable comments!

**Q1 Impact and novelty of theory**

**A1:** Conditional diffusion models (CDM) have emerged as a powerful generative model with diverse applications from image generation to control and RL[1, 2, 3].  In sharp contrast to abundant empirical successes, theoretical understanding of diffusion is limited, let alone theory of conditional diffusion. Our paper established the *first finite-sample statistical theory for training CDM* and *the first provable efficiency guarantee for CDM applied to optimizing reward*. We assume a practical encoder-decoder architecture for score matching and provide theoretical results that apply to both linear and nonparametric nonlinear reward (Theorem 3.5, 3.6), general distribution with mild regularity assumptions (Section 3.3), which covers the use of ReLU networks for training.

We summarize implications of our theory:

*a. Fidelity of generated samples:* Theorem 3.5 proves that conditional score matching network with architecture (2.8) accurately recovers any latent subspace structure in data.

*b. Trade-off between high target reward $a$ and large distribution-shift:* Theorem 3.6 shows that the average reward of generated samples is lower bounded by $a - Error(a)$. $Error(a)$ further decomposes into distribution shift penalties $\mathcal{E}_1$ (due to reward function estimation) and $\mathcal{E}_2$ (due to conditional score matching). It provides a comprehensive exposition of the interplay between reward signal and distribution-dependent factors.

*c.Low-dimensionality dependency:* $\mathcal{E}_1$ only depends on the subspace dimension $d$, free of the ambient dimension $D$. Moreover,  $\mathcal{E}_1$ turns out to be the distribution shift term commonly seen in offline bandit/RL, whereas they require prior knowledge of the latent space but our conditional diffusion automatically learns the latent space.

*d. Beyond linear settings:* We extend to nonparametric settings in Section 3.3, allowing general distributions and nonlinear rewards.

**Q2 Comparison between Alg.1, diffusion with classifier guidance or classifier free guidance**

**A2:** Alg. 1 is not an alternative to classifier(-free) guidance, but it is a simplification and a generalization of them for theoretical purposes. We will add the following extensive remark about this point in our final paper:

>Our work draws motivation from the guidance-based diffusion classifier which was originally designed for discrete labels-conditioned generation. We generalize it with provable guarantees to continuous reward by formulating Alg.1. Our analysis provides theoretical justification for reward-conditioned diffusion in control and RL [2,3], not limited to classification.
Notably, we provide the first theoretical guarantee for general conditional diffusion models. We show that CDM with score network as in (2.8) (FIgure 2(c)) guarantees generation fidelity in Theorem 3.5 for general light-tailed distribution with mild assumptions on score regularity.

>*Alg.1 and its analysis captures the essence of both classifier guidance and classifier free guidance methods.*  From a mathematical point of view, classifier guidance and classifier-free guidance share the same foundational objective, which is also the objective used in Alg. 1 — to estimate the conditional score $\nabla \log p_t (x_t | y)$. Accordingly, the foundational theoretical question is the same: understanding score matching and finite-sample distribution approximation, see Lemma E.1 in Appendix. Thus, one can view Alg. 1 as a meta algorithm and a simplification of these two guidance diffusion methods for elegance of mathematical analysis.  We provide more detailed discussion below:
>By the Bayes’ rule, it holds that
$$\nabla \log p_t\left(x_t \mid y\right)=\nabla \log p_t\left(x_t\right)+\nabla \log c_t\left(y \mid x_t\right). \qquad (\star)$$ Recall that Alg.1 directly learns the conditional score $\nabla \log p_t\left(x_t \mid y\right)$ on the LHS of ($\star$). The score network is trained on $(x, \hat{f}(x))$ pseudo-labelled by reward prediction $ \hat{f}$. Next we discuss its relation to classifier guidance and classifier-free guidance:
> + Classifier guidance [1] focuses on the RHS of ($\star$) and estimates $\nabla \log p_t\left(x_t\right)$ and $\nabla \log c_t\left(y \mid x_t\right)$ separately. Here $c_t$ is trained to take in noisy input $x_t$ and the time index $t$ to predict the label $y$. In contrast, Alg. 1 avoids the hassle of training a classifier/reward model with noisy inputs and additional input dimension $t$, and also offers theoretical cleanness.
> + Classifier-free guidance learns the same conditional score $\nabla \log p_t\left(x_t \mid y\right)$ as Alg.1 does, and uses a linear combination of conditioned and unconditioned scores for inference: $$\widetilde{s}_\theta(x, y, t)=(1+\eta) \widehat{s}(x, y, t)-\eta \widehat{s}(x, \emptyset, t).$$ When $\eta = 1$, Alg.1 is equivalent to classifier free guidance.

>We are not aware of any existing theory for classifier guidance and classifier-free guidance. Given that Alg.1 resembles yet simplifies both of them and keeps the essential spirit, we believe one can extrapolate from our theory to better understand these two methods.

**3 New Experiments on RL**

**A3:** To verify our theory beyond synthetic experiments and text-to-image (sections 4.1, 4.2), we conducted a new experiment using condition diffusion for offline RL, following [2]. See the pdf attached for details.
Further, we were able to match best known benchmarks of offline RL. Empirical observations match the our theory.

[1] "Classifier-free diffusion guidance." arXiv preprint arXiv:2207.12598 (2022).
[2] "Is conditional generative modeling all you need for decision-making?." arXiv preprint arXiv:2211.15657 (2022).(ICLR 2023 Oral)
[3]. "Adaptdiffuser: Diffusion models as adaptive self-evolving planners." arXiv preprint arXiv:2302.01877 (2023).

---

### Decision · Program_Chairs · 2023-09-21

**Decision:**

Accept (poster)

**Comment:**

The authors provided a theoretical justification of the reward-conditioned diffusion model. The main theoretical results are two-fold:

* 1) diffusion model can recover the linear subspace;
* 2) reward-conditioned diffusion model can adapt to the conditioned reward;

The results are technically sound and all of the reviewers leaned towards acceptance for this submission. I do hope the authors can include the discussion on the following two questions in the final version if possible:

* 1) what will happen if we mis-specify the latent dimension in our parameterization, say the latent dimension is $d$ but we specify it as $d'$ with $d' > d$ or $d' < d$?
* 2) why should we focus on the if we can if the expected reward on the conditional distribution match the target $y^*$, instead of directly bound some distances between the ground truth conditional distribution and learned conditional distribution?